# Development and Application of a United States wide correction for PM$_{2.5}$ data collected with the PurpleAir sensor

Karoline K. Barkjohn[1], Brett Gantt[2], & Andrea L. Clements[1]

[1] Office of Research and Development, U.S. Environmental Protection Agency 109 T.W. Alexander Drive Research Triangle Park, NC 27711

[2] Office of Air Quality Planning and Standards, U.S. Environmental Protection Agency, 109 T.W. Alexander Drive Research Triangle Park, NC 27711

*Correspondences to:* Karoline K. Barkjohn (Barkjohn.karoline@epa.gov)

**Abstract.** PurpleAir sensors, which measure particulate matter (PM), are widely used by individuals, community groups, and other organizations including state and local air monitoring agencies. PurpleAir sensors comprise a massive global network of more than 10,000 sensors. Previous performance evaluations have typically studied a limited number of PurpleAir sensors in small geographic areas or laboratory environments. While useful for determining sensor behavior and data normalization for these geographic areas, little work has been done to understand the broad applicability of these results outside these regions and conditions. Here, PurpleAir sensors operated by air quality monitoring agencies are evaluated in comparison to collocated ambient air quality regulatory instruments. In total, almost 12,000 24-hour averaged PM$_{2.5}$ measurements from collocated PurpleAir sensors and Federal Reference Method (FRM) or Federal Equivalent Method (FEM) PM$_{2.5}$ measurements were collected across diverse regions of the United States (U.S.), including 16 states. Consistent with previous evaluations, under typical ambient and smoke impacted conditions, the raw data from PurpleAir sensors overestimate PM$_{2.5}$ concentrations by about 40% in most parts of the U.S. A simple linear regression reduces much of this bias across most U.S. regions, but adding a relative humidity term further reduces the bias and improves consistency in the biases between different regions. More complex multiplicative models did not substantially improve results when tested on an independent dataset. The final PurpleAir correction reduces the root mean square error (RMSE) of the raw data from 8 µg m$^{-3}$ to 3 µg m$^{-3}$ with an average FRM or FEM concentration of 9 µg m$^{-3}$. This correction equation, along with proposed data cleaning criteria, has been applied to PurpleAir PM$_{2.5}$ measurements across the U.S. in the AirNow Fire and Smoke Map (fire.airnow.gov) and has the potential to be successfully used in other air quality and public health applications.

## 1    Introduction

Fine particulate matter (PM$_{2.5}$, the mass of particles with aerodynamic diameters smaller than 2.5 µm) is associated with a number of negative health effects (Schwartz et al., 1996;Pope et al., 2002;Brook et al., 2010). Short-term and long-term exposures to PM$_{2.5}$ are associated with increased mortality (Dominici et al., 2007;Franklin et al., 2007;Di et al., 2017). Even at low PM$_{2.5}$ concentrations, significant health impacts can be seen (Bell et al., 2007;Apte et al., 2015) and small increases of only 1-10 µg m$^{-3}$ can increase negative health consequences (Di et al., 2017;Bell et al., 2007;Grande et al., 2020). In addition

to health effects, PM$_{2.5}$ can harm the environment, reduce visibility, and damage materials and structures (Al-Thani et al., 2018;Ford et al., 2018). Understanding PM$_{2.5}$ at fine spatial and temporal resolutions can help mitigate risks to human health and the environment, but the high cost and complexity of conventional monitoring networks can limit network density (Snyder et al., 2013;Morawska et al., 2018).

Lower cost air sensor data may provide a way to better understand fine scale air pollution and protect human health. Air sensors are widely used by a broad spectrum of groups from air quality monitoring agencies to individuals. Sensors offer the ability to measure air pollutants at higher spatial and temporal scales than conventional monitoring networks with potentially less specialized operating knowledge and cost. However, concerns remain about air sensor data quality (Clements et al., 2019;Williams et al., 2019). Typically, air sensors require correction to become more accurate compared to regulatory monitors. A best practice is to locate air sensors alongside regulatory air monitors to understand their local performance and to develop corrections for each individual sensor (Jiao et al., 2016;Johnson et al., 2018;Zusman et al., 2020). For optical particulate matter (PM) sensors, correction procedures are often needed due to both the changing optical properties of aerosols associated with both their physical and chemical characteristics (Levy Zamora et al., 2019;Tryner et al., 2019), and the influence of meteorological conditions including temperature and relative humidity (RH) (Jayaratne et al., 2018;Zheng et al., 2018). In addition, some air sensors have out of the box differences and low precision between sensors of the same model (Feenstra et al., 2019;Feinberg et al., 2018). Although collocation and local correction may be achievable for researchers and some air monitoring agencies, it is unattainable for many sensor users and community groups due to lack of access and proximity to regulatory monitoring sites.

PurpleAir sensors are a PM sensor package consisting of two laser scattering particle sensors (Plantower PMS5003), a pressure-temperature-humidity sensor (Bosch BME280), and a WiFi-enabled processor that allows the data to be uploaded to the cloud and utilized in real-time. The low cost of outdoor PurpleAir sensors ($230-$260 U.S. dollars) has enabled them to be widely used with thousands of sensors publicly reporting across the U.S. Previous work has explored the performance and accuracy of PurpleAir sensors under outdoor ambient conditions in a variety of locations across the United States including in Colorado (Ardon-Dryer et al., 2020;Tryner et al., 2020a), Utah (Ardon-Dryer et al., 2020;Kelly et al., 2017;Sayahi et al., 2019), Pennsylvania (Malings et al., 2020), North Carolina (Magi et al., 2019), and in California where the most work has occurred to date (Ardon-Dryer et al., 2020;Bi et al., 2020;Feenstra et al., 2019;Mehadi et al., 2020;Schulte et al., 2020;Lu et al., 2021). Their performance has been explored in a number of other parts of the world as well including in Korea (Kim et al., 2019), Greece (Stavroulas et al., 2020), Uganda (McFarlane et al., 2021), and Australia (Robinson, 2020). Additional work has been done to evaluate their performance under wildland fire smoke impacted conditions (Bi et al., 2020;Delp and Singer, 2020;Holder et al., 2020), indoors (Wang et al., 2020b), and during laboratory evaluations (Kelly et al., 2017;Kim et al., 2019;Li et al., 2020;Mehadi et al., 2020;Tryner et al., 2020a;Zou et al., 2020a;Zou et al., 2020b). The performance of their dual Plantower PMS5003 laser scattering particle sensors has also been explored in a variety of other commercial and custom built sensor packages (He et al., 2020;Tryner et al., 2019;Kuula et al., 2019;Ford et al., 2019;Si et al., 2020;Zou et al., 2020b;Tryner et al., 2020b). Although not true of all types of PM$_{2.5}$ sensors, previous work with PurpleAir sensors and other

models of Plantower sensors have shown that the sensors are precise, with sensors of the same model measuring similar $PM_{2.5}$ concentrations (Barkjohn et al., 2020a;Pawar and Sinha, 2020;Malings et al., 2020). However, extensive work with PurpleAir and Plantower sensors has often shown deficiencies in the accuracy of the measurement resulting in the need for correction. A number of previous corrections have been developed; however, they are typically generated for a specific region, season, or condition, and little work has been done to understand how broadly applicable they are (Ardon-Dryer et al., 2020;Magi et al., 2019;Delp and Singer, 2020;Holder et al., 2020;Tryner et al., 2020a;Robinson, 2020). Although location-specific and individual sensor corrections may be ideal, the high precision suggests that a single correction across PurpleAir sensors may be possible. This is especially important since having multiple corrections can make it difficult for many sensor users to know which correction is best for their application.

In this work, we develop a U.S.-wide correction for PurpleAir data which increases accuracy across multiple regions making it accurate enough to communicate the Air Quality Index (AQI) to support public health messaging. We use onboard measurements and information that would be available for all PurpleAir sensors, even those in remote areas far from other monitoring or meteorological sites.

## 2    Data Collection

### 2.1    Site identification

Data for this project came from 3 sources: 1) PurpleAir sensors sent out by EPA for collocation to capture a wide range of regions and meteorological conditions, 2) privately operated sensor data volunteered by state, local and tribal (SLT) air monitoring agencies independently operating collocated PurpleAir sensors, and 3) publicly available sensors located near monitoring stations and confirmed as true collocation by air monitoring agency staff. In order to identify publicly available collocated sensors, in August of 2018, a survey of sites with potentially collocated PurpleAir sensors and regulatory $PM_{2.5}$ monitors was performed by identifying publicly available PurpleAir sensor locations within 50 meters of an active EPA Air Quality System (AQS) site reporting $PM_{2.5}$ data in 2017 or 2018. The 50-meter distance was selected because it is large enough to cover the footprint of most AQS sites and small enough to exclude most PurpleAir sensors in close proximity, but not collocated with, an AQS site. From a download of all active AQS $PM_{2.5}$ sites and PurpleAir sensor locations on August 20, 2018, 42 unique sites were identified in 14 states (data from additional states were available from sensors sent out by EPA and privately operated sensors). From this list of public PurpleAir sensors potentially collocated with regulatory $PM_{2.5}$ monitors, we reached out to the appropriate SLT air monitoring agency to understand if these units were operated by the air monitoring agency and their interest in partnering in this research effort. If we could not identify the sensor operator of these 42 sensors, or if the sensor was not collocated at the air monitoring station, the sensor was not used in this analysis.

Much past work using public data from PurpleAir has used public sensors that appear close to a regulatory station on the map (Ardon-Dryer et al., 2020;Bi et al., 2020). However, there is uncertainty in the reported location of PurpleAir sensors as this is specified by the sensor owner. In some cases, sensors may have the wrong location. Known examples include owners who forgot to update the location when they moved, take the sensor inside for periods to check their indoor air quality, or

specifically choose an incorrect location to protect their privacy. In addition, without information on local sources of PM$_{2.5}$, it can be unclear how far away is acceptable for a "collocation" since areas with more localized sources will need to be closer to the reference monitor to experience similar PM$_{2.5}$ conditions. By limiting this work to true collocations operated by air monitoring agencies, we eliminate one source of uncertainty. We can conclude that the PurpleAir errors measured in this work are not due to poor siting or localized sources and can focus on other variables that influence error (e.g. RH).

When EPA provided PurpleAir sensors to air monitoring agencies, EPA suggested that they be deployed with similar siting criteria as regulatory monitors. Some sites had space and power limitations to consider but trained technicians sited sensors allowing adequate unobstructed airflow. In many cases, sensors were attached to the top rung of the railings at the monitoring shelters where they were within a meter or so of other inlet heights and within 3 meters or so of the other instrument inlets.

In total, 53 PurpleAir sensors at 39 unique sites across 16 states were ideal candidates and were initially included in this analysis with data included from September 2017 until January 2020. The supplement contains additional information about each AQS site (Table S1) and each individual sensor (Table S2).

### 2.1.1 Subsetting the Iowa dataset

Initially, there were 10,907 pairs of 24-hr averaged collocated data from Iowa which was 55% of the entire collocated dataset. In order to better balance the dataset among the states, and to avoid building a correction model that is weighted too heavily towards the aerosol and meteorological conditions experienced in Iowa, the number of days from Iowa was reduced to equal the size of the California dataset, the state with the next largest amount of data (29% of the entire collocated dataset). When reducing the Iowa dataset, the high concentration data were preserved. Although high 24-hour PM$_{2.5}$ averages occurred less frequently, they may have larger public health consequences and be of greater interest to communities. To preserve more of the high concentration data, the Iowa PurpleAir PM$_{2.5}$ data were split into 10 bins from 0-64 µg m$^{-3}$ by 6.4 µg m$^{-3}$ increments. Since there were less data in the higher concentration bins, all data in bins 6-10 ($\geq$25 µg m$^{-3}$) were included and an equal number of randomly selected data points was selected from each of the other 4 bins (N=649). The subset and full complement of Iowa data were compared visually and the distributions of the temperature and RH for both datasets were similar (Figure S1) with a similar range of dates represented from September 2017 until January of 2020.

## 2.2 Air monitoring instruments and data retrieval

### 2.2.1 PurpleAir sensors

The PurpleAir sensor contains two Plantower PMS5003 sensors, labelled as channel A and B, that operate for alternating 10-second intervals and provide 2-minute averaged data (prior to May 30, 2019, this was 80-second averaged data). Plantower sensors measure 90-degree light scattering with a laser using 680±10 nm wavelength light (Sayahi et al., 2019) and are factory calibrated using ambient aerosol across several cities in China (Malings et al., 2020). The Plantower sensor reports estimated mass of particles with aerodynamic diameters <1 µm (PM$_1$), <2.5 µm PM$_{2.5}$, and <10 µm (PM$_{10}$). These values are reported in two ways, labelled as cf_1 and cf_atm, in the PurpleAir dataset which match the "raw" Plantower outputs. PurpleAir

previously had these cf_1 and cf_atm column labels flipped in the data downloads (Tryner et al., 2020a), but for this work we have used the updated labels. The two data columns have a [cf_atm]/[cf_1] = 1 relationship below roughly 25 µg m$^{-3}$ (as reported by the sensor), and then transitions to a 2/3 ratio at higher concentration ([cf_1] concentrations are higher). The cf_atm data, displayed on the PurpleAir map, is the lower measurement of PM$_{2.5}$ and will be referred to as the "raw" data in this paper when making comparison between initial and corrected datasets. In addition to PM$_{2.5}$ concentration data, the PurpleAir sensors also provide the count of particles per 0.1 liter of air above a specified size in µm (i.e. >0.3, >0.5, >1.0, >2.5, >5.0, >10 µm); however, these are actually calculated results as opposed to actual size bin measurements (He et al., 2020).

When a PurpleAir sensor is connected to the internet, data is sent to PurpleAir's data repository on ThingSpeak. Users can choose to make their data publicly viewable (public) or control data sharing (private). Agencies with privately reporting sensors provided application programming interface (API) keys so that data could be downloaded. PurpleAir PA-II-SD models can also record data offline on a microSD card; however, these offline data appeared to have time stamp errors from internal clocks that drift without access to the frequent time syncs available with access to WiFi so they were excluded from this project. Data were downloaded from the ThingSpeak API using Microsoft PowerShell at the native 2-minute or 80-second time resolution and were saved as csv files that were processed and analysed in R (R Development Core Team, 2019).

### 2.2.2 Federal Reference Method (FRM) and Federal Equivalent Method (FEM) PM$_{2.5}$

24-hour averaged PM$_{2.5}$ reference data was downloaded for the 39 collocation sites from the AQS database on February 20, 2020 for both FRM and FEM regulatory monitors. Collocation data was collected from 9/28/2017 (the date at which the first collocated PurpleAir sensor was installed among the sites used in this study) through to the most recent quality assured data uploaded by each SLT agency (nominally 1/13/20). The 24-hour averages represent concentrations from midnight to midnight local standard time from either a single 24-hour integrated filter-based FRM measurement or an average of at least 18 valid hours of continuous hourly-average FEM measurements (75% data completeness). In our analysis, we included sample days flagged or concurred-upon as exceptional events to ensure that days impacted by wildfire smoke or dust storms with very high PM$_{2.5}$ concentrations would be considered in the correction.

National Ambient Air Quality Standards (NAAQS) set a 24-hour average standard for PM$_{2.5}$ so the PurpleAir sensor and FRM or FEM comparison used daily averaged data (midnight to midnight). This also allows for comparison of PurpleAir data to both FRM and FEM PM$_{2.5}$ measurements, which are expected to provide near-equivalent measurements at this time averaging interval. The use of 24-hour averages also benefits from the 1) improved inter-comparability between the different FEM instruments (Zikova et al., 2017), and 2) avoidance of the variability in short-term (1-minute to 1-hour) pollutant concentrations compared to longer term averages as used in the NAAQS (Mannshardt et al., 2017).

The dataset was comprised of data from 21 BAM 1020s or 1022s, 19 Teledyne T640 or T640xs, and 5 TEOM 1405s or 1400s. Sixteen sites had FRM measurements. After excluding part of the Iowa dataset BAM1020's provided the most 24-hr averaged points followed by the T640 and T640x, and the RP2025 (Figure S2).  1/5 of the data came from FRM measurements while the rest came from FEMs (Figure S3). If daily measurements were collected using two methods both points were included in the analysis.

# 3 Quality Assurance

## 3.1 FRM and FEM Quality Assurance

The accuracy of the FRM and FEM measurements was considered. In total, Federal Reference Method (FRM) data was used from 13 organizations. The accuracy of these measurements was evaluated using the FRM performance assessments (U.S. EPA, 2020b). They were evaluated using the FRM/FRM precision and bias, the average field blank weight, and the monthly precision. The performance of the FEM monitors was evaluated using the $PM_{2.5}$ continuous monitor comparability assessments (U.S. EPA, 2020a). FEM measurements are compared to simultaneous Federal Reference Method (FRM) measurements. Linear regression is used to calculate a slope, intercept (int), and correlation (R) and the FEM/FRM ratio is also computed. Based on data quality objectives, the slope should be between 0.9 and 1.1, intercept between -2 and 2, correlation should be between 0.9 and 1, and the ratio should be within 0.9 and 1.1. The most recent 3 years of available data was used to evaluate each monitor.

Performance data was only available for 10 of the 13 collection agencies (77%, Table S3). All available agencies met the FRM/FRM precision goals. All but one state show negative FRM bias suggesting organization reported FRM $PM_{2.5}$ is biased low by 1-22%. Four of the agencies (40%) only marginally fail the ≤10% bias criteria with bias from -10.1% to -11%. The one organization with more significant bias (-22%) is driven by the difference in a single FRM measurement pair. All sites typically have acceptable field blank weights and monthly average precision within 30%. The performance of all FRM measurements are acceptable for use in developing the PurpleAir U.S.-wide correction.

Of the 46 unique FEM monitors, comparability assessments were only available for 24 monitors (51%, Tables S4,S5). All slopes were within the acceptable range. One intercept was slightly outside the acceptable range (2.35) and 3 correlations were slightly below the acceptable limit (0.86-0.89), however these values have been considered acceptable for this use. Of greater concern is that 10 FEMs had ratios greater than 1.1 up to 1.3 (41% of monitors) and these were all Teledyne T640 or T640x devices (Figure S4). The data from the T640 and T640x make up about 20% of the total dataset and excluding them would reduce the diversity of the dataset. Since these monitors are frequently used for regulatory applications, the performance of all FEM measurements has been considered acceptable for use in developing the PurpleAir U.S.-wide correction.

## 3.2 PurpleAir Quality Assurance & Data Cleaning

### 3.2.1 PurpleAir averaging

The 2-minute (or 80-second) $PM_{2.5}$ data were averaged to 24-hours (representing midnight to midnight local standard time). A 90% data completeness threshold was used based on channel A, since both channels were almost always available together (i.e. 80-second averages required at least 0.9*1080 points before 5/30/2019 or 2-minute averages required at least 0.9*720 points after 5/30/2019). This methodology ensured that the averages used were truly representative of daily averages reported by regulatory monitors. A higher threshold of completeness was used for the PurpleAir data as it likely has more error than FEM or FRM measurements.

### 3.2.2 PurpleAir Temperature and RH errors

For correction model development, it was important to start with the most robust dataset possible. In the 2-minute or 80-second data, occasionally, an extremely high temperature (i.e. 2147483447) or an extremely low temperature (i.e. -224 or -223) was reported, likely due to electrical noise or a communication error between the temperature sensor and the PurpleAir microcontroller. The high error occurred in 24 of 53 sensors but occurred infrequently (34 instances in ~$10^7$ points total) while the low error impacted only 2 sensors (1% of the full dataset). Temperature values above 540°C (1000°F) were excluded before calculating daily averages since temperature errors were extreme and easily detected above this level. After excluding these values reasonable 24-hr averaged temperature values were generated (min= -25°C, max=44°C). Future work may wish to apply a narrower range of acceptable temperature ranges, accounting for typical ambient conditions and the potential for increased heat build-up inside sensors (discussed further in section 4.1). Similarly, the RH sensor occasionally read 255%; this problem was experienced by each sensor at least once but still occurred infrequently (1083 points out of ~$10^7$ total). No other values were found outside 0-100% in the 2-minute or 80-second data before averaging. These points were removed from the analysis before 24-hr averaging.

Missing temperature or RH impacted only 2% of the dataset (184 points) with 8 sensors having one to four 24-hr averages with missing temperature or RH. One sensor, WI4, had 167 days (90%) without temperature data. Most of the available temperature data was recorded in the first few weeks of operation. It is unclear what caused the temperature data to be missing. All 184 points were missing temperature but only 17 were also missing RH (0.2% of full dataset).

### 3.2.3 Comparison of A and B channels

The two Plantower sensors within the PurpleAir sensor (channels A and B) can be used to check the consistency of the data reported. All comparisons in this work have occurred at 24-hour averages. Anecdotal evidence from PurpleAir suggests some disagreements may be caused by spiders, insects, or other minor blockages that may resolve on their own. Data cleaning procedures were developed using the typical 24-hr averaged agreement between the A and B channels expressed as percent error (Eq. 1).

$$\text{24-hr percent difference} = \frac{(A-B)*2}{(A+B)}$$

(1)

Where A and B are the 24-hr average PM$_{2.5}$ cf_1 concentrations from the A and B channels. 24-hour averaged data points with percent differences larger than two standard deviations (2sd=61%) were flagged for removal. At low concentrations, where a difference of a few µg m$^{-3}$ could result in a percent error greater than 100%, an absolute concentration difference threshold of 5 µg m$^{-3}$, previously proposed by Tryner et al. (2020), was effective at removing questionable observations but was not appropriate at higher concentrations where a 5 µg m$^{-3}$ difference was more common but only represents a small percent difference. Therefore, data were cleaned using a combination of these metrics; data were considered valid if the difference between channels A and B was less than 5 µg m$^{-3}$ or 61%.

As illustrated in Figure 1, 24-hour averaged $PM_{2.5}$ concentrations reported by channels A and B generally agree exceptionally well (e.g., AZ1 sensor). However, our observations suggest there are some sensors where the two channels show a systematic bias out of the box (e.g., AK3 is the most apparent example), one channel reports zeros (e.g., CA4), or when reported concentrations do not match for a time but then recover (e.g., KS2). For this work, 24-hour averages were excluded from the dataset when the PurpleAir A and B channel [cf_1] $PM_{2.5}$ concentrations differed by more than 5 µg m$^{-3}$ and 61%. This resulted in removal of 0-47% of the data from individual sensors (Figure 1, Table S6) and 2.1% of the data in the full dataset. Most sensors had little to no removal (N= 48, <10% removed); 5 sensors had 10% to 47% removed (AK2, AK3, CA4, CA7, WA5). Of these sensors, 3 had average channel differences of more than 25% (27-45%), after applying 24-hour AB channel comparison removal criteria (AK3, CA7, WA5). These sensors, representing 3% of the dataset, were removed from further analysis because of the additional error they could add into the correction model building. In some cases, additional quality assurance checks on either the part of PurpleAir or the purchaser could identify problem sensors before they were deployed but this would not catch all issues as many occurred after sensors had operated for a while. In addition, when errors occur between channels some agencies cleaned the sensors with canned air or vacuums as recommended by PurpleAir.

Previous work with PurpleAir sensors excluded sensors for poor Pearson correlations but our work shows that a more targeted approach may be more efficient for ensuring good quality data. Previous work with PurpleAir sensors reported that 7 of 30 sensors (23%) were defective out of the box and exhibited low Pearson correlations (r < 0.7) in a laboratory evaluation (Malings et al., 2020). Ten of 53 sensors (19%) in our study had r<0.7 at 24-hour averages (i.e. AK2, CA3, CA4, CA7, IA10, KS2, WA2, WA3, WA5, WI2); only two of these were removed due to large average percent differences after removing outliers where A and B channels did not agree (i.e. WA5, CA7). Six of these 10 sensors had ≤4% of the data removed by data cleaning steps and their Pearson correlation, on 24-hour averages, improved to ≥0.98 (from r < 0.7) suggesting that the low correlation was driven by a few outlier points. Some sensors with low initial Pearson correlations had high Spearman correlations (range: 0.69 to 0.98); this suggests, again, that the low performance was due to a few outlier points. These results highlight that sensors may fail checks based on Pearson correlation or overall percent difference thresholds due to only a small fraction of points often making them poor indicators of overall sensor performance. The removal of outliers after comparing the A and B channels can greatly improve agreement between sensors and between sensors and reference instruments.

### 3.2.4    Importance of PurpleAir data cleaning procedures

This work did not seek to optimize data cleaning procedures to balance data retention with data quality, instead it focused on generating a best-case dataset from which to build a model. However, the removal of outlier points based on the difference between the A and B channels appears to reduce the errors most strongly (Supplement section 3, Table S7) when compared to removing incomplete daily averages or removing problematic sensors. Since both channels are needed for comparison, it makes sense to average the A and B channels to improve the certainty on the measurement. The data completeness control provides less benefit and may not be needed for all future applications of these correction methods. In

addition, sensors with systematic offsets were uncommon and did not largely impact the overall accuracy, so the A and B channel comparison on the 24-hour averaged data points (e.g. 5 µg m$^{-3}$ and 61%) may be sufficient.

## 3.3    Data summary

After excluding poorly performing sensors (N=3), 50 PurpleAir sensors were used in this analysis. These sensors were located in 16 states across 39 sites (Figure 2, Table 1). Some sites had several PurpleAir sensors running simultaneously (N=9) and one ran multiple sensors in series (i.e. one sensor failed, was removed, and another sensor was put up in its place). Some states had more than two years of data while others had data from a single week or season. Median state-by-state PM$_{2.5}$ concentrations, as measured by the FRM or FEM, ranged from 4-10 µg m$^{-3}$. A wide range of PM$_{2.5}$ concentrations was seen across the dataset with a maximum 24-hour average of 109 µg m$^{-3}$ measured in California; overall the median PM$_{2.5}$ concentration of the dataset was 7 µg m$^{-3}$ (inter quartile range: 5-11 µg m$^{-3}$, average(sd): 9(5) µg m$^{-3}$). These summary statistics were calculated after selecting a random subset of the Iowa data. The median of the Iowa dataset increased from 7 to 10 µg m$^{-3}$ after subsetting since more of the high concentration data was conserved. Sensors were located in several U.S. climate zones (NOAA, 2020;Karl and Koss, 1984) resulting in variable temperature and RH ranges (Figure S5). There was limited data above 80% RH as measured by the PurpleAir RH sensor, which, as discussed in the following section, is known to consistently report RH lower than ambient.

## 4    Model Development

### 4.1    Model input considerations

To build a data correction model that could easily be applied to all PurpleAir sensors, only data reported by the PurpleAir sensor (or calculated from these parameters) were considered as model inputs. The 24-hour FRM or FEM PM$_{2.5}$ concentrations were treated as the independent variable (plotted on x-axis) allowing the majority of error to reside in the PurpleAir concentrations. We considered a number of redundant parameters (i.e. multiple PM$_{2.5}$ measurements, multiple environmental measurements) and we considered a number of increasingly complex models where parameters that were not strongly correlated were included as additive terms with coefficients or where they were multiplied with each other to form more complex models accounting for collinearity. Increasingly complex models were evaluated based on the reduction in root mean squared error (RMSE, Eq. S1). Subsequently, several of the best performing model forms were validated using withholding methods as described in the next section.

In a multiple linear regression, all independent variables should be independent; however, much previous work has used models that incorporate additive temperature, RH, and dewpoint terms that are not independent (Magi et al., 2019;Malings et al., 2020). We have not considered these models and have considered models with interaction terms (i.e. RH*T*PM$_{2.5}$) to account for inter-dependence between terms instead. Strong correlations (r ≥ ±0.7) are shown between the 24-hr averaged FEM or FRM PM$_{2.5}$, PurpleAir estimated PM$_{2.5}$ (cf_1and cf_atm), and each binned count (Figure S6). Since the binned counts include all particles greater than a certain size, we also consider the correlation between the delta of each bin (e.g. particles >0.3 µm – particles >0.5 µm=particles 0.3-0.5 µm). The delta bin counts were still moderately to strongly correlated (r=0.6-1)

with the weakest correlation seen between the smallest and largest bins (Figure S7). Moderate correlations (r = ±0.4-0.6) are seen between temperature, RH, and dewpoint. Weak correlations (r ≤ ±0.2) are seen between the PM variables (i.e. $PM_{2.5}$ and bin variables) and environmental variables (i.e. temperature, RH, and Dewpoint). The correlation between variables was considered when considering model forms.

For PM, we considered $PM_{2.5}$ concentrations from both the [cf_1] and [cf_atm] data columns as model terms. Previous work has found different columns to be more strongly correlated under different conditions (Barkjohn et al., 2020a;Tryner et al., 2020a). Previous studies have suggested that the binned particle count data from the Plantower is more effective at estimating $PM_{2.5}$ concentrations than the reported $PM_{2.5}$ concentration data from the newer Plantower PMSA003 sensor (Zusman et al., 2020). However, the bins are highly correlated and so multilinear equations with additive terms representing

the bins were not considered.

Temperature, RH, and dewpoint, as calculated from the reported temperature and RH data, were also considered based on previous studies (Malings et al., 2020). Dew point was considered since past work has shown that dewpoint can, in some cases, explain error unexplained by temperature or RH (Mukherjee et al., 2019;Malings et al., 2020). Pressure was not a reported variable for 10% of the dataset and was therefore not considered as a possible correction parameter.

Both linear and non-linear RH terms were considered. Previous studies often used a nonlinear correction for RH as opposed to a correction that changes linearly with RH (Stampfer et al., 2020;Tryner et al., 2020a;Kim et al., 2019;Malings et al., 2020;Zheng et al., 2018;Lal et al., 2020). A nonlinear RH model was tested by adding a $RH^2/(1-RH)$ term (see Eq. 3) similar to what has been used in past work for Plantower sensors and other light scattering measurements (Tryner et al., 2020a;Malings et al., 2020;Chakrabarti et al., 2004;Zheng et al., 2018;Zhang et al., 1994;Day and Malm, 2001;Soneja et al.,

2014;Lal et al., 2020;Barkjohn et al., 2020b). In Eq. 2, PA is the PurpleAir $PM_{2.5}$ data and $PM_{2.5}$ is the concentration provided by the collocated FRM or FEM.

$$PA = s_1 * PM_{2.5} + s_2 \frac{RH^2}{(1-RH)} * PM_{2.5} + s_3 * \frac{RH^2}{(1-RH)} + i$$

(2)

It is important to note that the meteorological sensor in the PurpleAir sensor is positioned above the particle sensors

nestled under the PVC cap resulting in temperatures that are higher (2.7 to 5.3 °C) and RH that is drier (9.7% to 24.3%) than ambient conditions (Holder et al., 2020;Malings et al., 2020). In addition, these internal measurements have been shown to be strongly correlated with reference temperature and RH measurements with high precision (Holder et al., 2020;Tryner et al., 2020a;Magi et al., 2019). The well characterized biases and strong correlations between PurpleAir and ambient meteorological parameters mean that the coefficients using these terms in a correction equation account for the differences between the

ambient and PurpleAir measured meteorology. Although not as accurate as the reference measurements, the PurpleAir temperature and RH measurements are good candidates for inclusion in a linear model because they are well correlated with reference measurements and may more closely represent the particle drying that occurs inside the sensor. In addition, using

onboard measurements and information that would be available for all PurpleAir sensors allows us to gather corrected air quality data from all PurpleAirs, even those in remote areas far from other air monitoring or meteorological sites.

### 4.1.1    Selecting models

RMSE was used to determine the best models of each increasing complexity moving forward (Table 2). The $PM_{2.5}$ [cf_1] data resulted in less error than the [cf_atm] (Figure 3) across all model forms (Table 2). The modest change in RMSE reflects the fact that only 3.8% of the dataset has FRM or FEM $PM_{2.5}$ concentrations greater than 20 µg m$^{-3}$ which is where these two data columns exhibit a different relationship. Previous work with Plantower sensors in the U.S. has shown nonlinearity at higher concentrations >10-25 µg m$^{-3}$, which we do not see, which appears due to the use of the [cf_atm] data in the previous work (Stampfer et al., 2020;Kelly et al., 2017;Malings et al., 2020).

The next most complex model considered adding a single additive term representing the meteorological variables. Including an additive, linear RH term to a model already including the [cf_1] PurpleAir $PM_{2.5}$ data yielded the lowest error (RMSE=2.52) with dewpoint reducing error less than temperature (+D RMSE=2.86 µg m$^{-3}$, +T RMSE=2.84 µg m$^{-3}$). Since the linear model with RH has the best performance of these combinations, it will be further considered in the next section.

As in previous studies with Plantower sensors, the PurpleAir sensors appear to overestimate $PM_{2.5}$ concentrations at higher RH (Tryner et al., 2020a;Magi et al., 2019;Malings et al., 2020;Kim et al., 2019;Zheng et al., 2018). Overestimation was observed in our dataset before correction as shown in Figures S8 and S9 with overestimation increasing between 30% and 80%. There are few 24-hr averages above 80% RH so there is more uncertainty in the relationship above that level although it appears to level off. However, the RH$^2$/(1-RH) had higher error than using the same equation with just RH (nonlinear RH: RMSE=2.86 µg m$^{-3}$, +RH term: RMSE=2.52 µg m$^{-3}$) so this model form will not be used moving forward. This result suggests that there may be other variations in aerosol properties and meteorology in this nationwide dataset which are not well captured just by considering hygroscopicity. This term may be more significant in localized areas with high sulfate and nitrate concentrations where aerosol hygroscopicity plays an important role.

More complex models, which add terms to account for interactions between environmental conditions were also considered (Table 2 rows 6-9).  Lower error was observed for the +D*T (RMSE=2.51 µg m$^{-3}$) compared to other models of similar complexity and slightly lower error was observed when adding RH as well (+RH*T*D RMSE=2.48 µg m$^{-3}$). Adding interaction terms between $PM_{2.5}$ and the environmental conditions reduced error even more with PM*RH having the lowest error for a single interaction term (RMSE=2.48 µg m$^{-3}$), although the error was similar to the +RH*T*D model. Adding a second interaction term for T explains slightly more error (PM*RH*T RMSE=2.46 µg m$^{-3}$) and the most error is explained by the most complex model (PM*RH*T*D RMSE=2.42 µg m$^{-3}$). These best performing models will be further considered in the next section.

### 4.1.2    Models considered

Based on this analysis, the seven models with the lowest RMSE were explored further. In those equations, shown below, PA represents the PurpleAir $PM_{2.5}$ [cf_1] data, $PM_{2.5}$ represents the $PM_{2.5}$ concentration provided by the collocated

FRM or FEM, $s_1$-$s_7$ are the fitted model coefficients, i is the fitted model intercept, and RH and T represent the RH and temperature as measured by the PurpleAir sensors.

1. Simple Linear Regression

$$PA = s_1*PM_{2.5} + i$$

(3)

2. Multilinear with an additive RH term

$$PA = s_1*PM_{2.5} + s_2*RH + i$$

(4)

3. Multilinear with additive T and D interaction terms

$$PA = s_1*PM_{2.5} + s_2*D + s_3*T + s_4*D*T + i$$

(5)

4. Multilinear with additive and multiplicative terms using RH, T and D

$$PA = s_1*PM_{2.5} + s_2*RH + s_3*RH*PM_{2.5} + i$$

(6)

5. Multilinear with additive and multiplicative terms using RH and $PM_{2.5}$

$$PA = s_1*PM_{2.5} + s_2*RH + s_3*RH*PM_{2.5} + i$$

(7)

6. Multilinear with additive and multiplicative terms using T, RH, and $PM_{2.5}$

$$PA = s_1*PM_{2.5} + s_2*RH + s_3*T + s_4*PM_{2.5}*RH + s_5*PM_{2.5}*T + s_6*RH*T + s_7*PM_{2.5}*RH*T + i$$

(8)

7. Multilinear with additive and multiplicative terms using T, RH, D and $PM_{2.5}$

$$PA = s_1*PM_{2.5} + s_2*RH + s_3*T + s_4*D + s_5*PM_{2.5}*RH + s_6*PM_{2.5}*T + s_7*T*RH + s_8*PM_{2.5}*D + s_9*D*RH + s_{10}*D*T$$
$$+s_{11}*PM_{2.5}*RH*T + s_{12}*PM_{2.5}*RH*D + s_{13}*PM_{2.5}*D*T + s_{14}*D*RH*T + s_{15}*PM_{2.5}*RH*T*D + i$$

(9)

## 5    Model Evaluation

### 5.1    Model validation methods

Building the correction model based on the full dataset could lead to model overfitting so two different cross-validation structures were used: 1) "leave-out-by-date" (LOBD) and 2) "leave-one-state-out" (LOSO). For the LOBD model validation method, the project time period was split into 4-week periods with the last period running just short of 4 weeks (24 days). Each period contained between 179 and 2,571 24-hr data points with typically more sensors running continuously during later chunks as more sensors were deployed and came online over time. Thirty periods were available in total and, for each test-train set, 27 periods were used to train the correction model while three periods were selected to test the correction model. Models were generated for all 27,000 combinations of test data. For the LOSO model validation method, the correction model was built based on sensors from all but one state and then the model was tested on data from the withheld state. This resulted in 16 unique models since there are 16 states represented in this dataset. The LOSO method is useful for understanding how well the proposed correction may work in geographic areas that are not represented in our dataset. The performance of each correction method on the test data was evaluated using the RMSE, the mean bias error (MBE), the mean absolute error (MAE), and the Spearman correlation ($\rho$). Equations for these statistics are provided in the SI (Section S1). To compare statistical difference between errors, t-tests were used to compare normally distributed datasets (as determined by Shapiro-Wilk) and Wilcoxon Signed Rank tests were used for nonparametric datasets with a significance value of 0.05. Both tests were used in cases where results were marginal.

### 5.2    Model evaluation

Figure 4. Performance statistics including mean bias error (MBE) and mean absolute error (MAE) are shown by correction method (0-7), where each point in the boxplot is the performance for either a 12-week period excluded from correction building ("LOBD"), or a single state excluded from correction building ("LOSO").

shows the performance of the raw and corrected PurpleAir $PM_{2.5}$ data using the seven proposed correction models for datasets of withheld dates (LOBD) or states (LOSO). Both MBE, which summarizes whether the total test dataset measures higher or lower than the FRM or FEM measurements, and MAE, which summarizes the error on 24-hr averages, are shown with additional statistics and significance testing shown in the supplement (Table S8, S9). Large reductions in MAE, and MBE are seen when applying a linear correction (Eq. 3). Using LOBD, the MBE across withholding runs drops significantly from 4.2 to 0 µg m$^{-3}$ with a similar significant drop, from 2.8 to 0 µg m$^{-3}$, for LOSO withholding as well. This is a large improvement considering the average concentration in the dataset is 9 µg m$^{-3}$. When applying an additive RH term (+RH), the MAE improves significantly by 0.3 µg m$^{-3}$ for LOBD and by 0.4 for LOSO. Median LOSO and LOBD MBE do not change significantly. The inter quartile range (IQR) improves for both metrics and withholding methods showing that models typically have more consistent performance across withheld datasets. Overall, the additive RH correction model improves performance over the linear correction.

Increasing the complexity of the model (Eq. 5-9) shows similar performance to the additive RH model with no further improvements in MBE or RMSE for LOSO withholding. When using the multiplicative RH model, the MAE changes significantly (t-test, Wilcoxon-test) however, the median values does not largely change (1.6 µg m$^{-3}$). However, because this dataset contains limited high concentration with a limited range of RH experienced at higher concentration, there is greater uncertainty in how this model would perform when extrapolated into such conditions. Therefore, the additive RH model was used moving forward. However, future work should look at larger datasets to understand whether a multiplicative RH correction is more appropriate. Further, model coefficients become more variable for more complex models depending on the dataset that is excluded suggesting that individual states or short time periods may be driving some of the coefficients in the more complex models (Table S10). In addition, since temperature and relative humidity are moderately correlated, they may be providing very similar information to the model. Since more complex models do not improve median MAE, MBE, or RMSE for LOSO withholding and since more complex models will be applicable for a narrower window of conditions, the additive RH correction was selected as being most robust.

**5.3    Selected correction model**

In the end, the additive RH model (Eq. 4) seems to optimally summarize a wide variety of data while reducing error (MAE) compared to a simple linear correction. The following correction model (Eq. 10) was generated for the full dataset where PA is the average of the A and B channels from the higher correction factor (cf_1) and RH is in percent.

$$PM_{2.5} = 0.524*PA_{cf\_1} - 0.0862*RH + 5.75$$

(10)

This work indicates that only an RH correction is needed to reduce the error and bias in the nationwide dataset. Some previous single site studies found temperature to significantly improve their $PM_{2.5}$ prediction as well (Magi et al., 2019;Si et al., 2020). Humidity has known impacts on the light scattering of particles; no similar principle exists for explaining the

influence of temperature on particle light scattering. Instead, the temperature factor may help account for some local seasonal or diurnal patterns in aerosol properties within smaller geographical areas. These more local variations may be why temperature does not substantially reduce error and bias in the nationwide dataset. More work should be done to better understand this influence. These previous models also did not include a term accounting for the collinearity between temperature and relative humidity that may have been present. Figure 5 shows the residual error in each 24-hour corrected PurpleAir $PM_{2.5}$ measurement compared with the temperature, RH, and FRM or FEM $PM_{2.5}$ concentrations. Error has been reduced compared to the raw dataset (Figures S8, and S9) and is unrelated to temperature, RH, and $PM_{2.5}$ variables. Some bias at very low temperature < -12°C and potentially high concentration (> 60 µg m$^{-3}$) may remain, but more data are needed to further understand this relationship.

### 5.3.1 The influence of FEM and FRM type

We briefly considered whether the use of both FEM and FRM measurements influenced these results. When sub-setting the data to develop models using the 24-hr averaged $PM_{2.5}$ data from only the FEM versus only the FRM, only the coefficient for the PA slope term changed. The coefficient was slightly larger for FEM measurement (0.537) and smaller for FRM measurements (0.492). Although the coefficients are significantly different (p<0.05) they are within 10% leading to little difference in the interpretation of PurpleAir $PM_{2.5}$ measurements. We briefly considered whether the FEM coefficient was driven by the T640s and found that if we build this model excluding all T640 and T640x data, it is not significantly different (0.53). Concerns about error between different types of FEM measurements cannot be explored using this dataset. Further, FEM instruments are not randomly distributed across the U.S. but rather clustered at sites operated by the same air agency. Future work and a more concerted effort may be needed to explore this issue. Overall, the accuracy of all these FEM and FRM methods have been determined accurate enough for regulatory purposes and so we have used all to determine our U.S.-wide correction. Although FRM measurements are the gold standard, using only FRM measurements would have severely limited our dataset. In addition, the use of FEM measurements will be important in future work to explore the performance of this model correction at higher time resolutions. At higher time resolutions, the noise and precision between different FEMs may impact perceived performance and future work should further explore this.

### 5.3.2 Error in corrected data by region

The performance of the selected model is summarized by region. Sites were first divided by the National Oceanic and Atmospheric Administration's (NOAA) U.S. Climate Regions (NOAA, 2020;Karl and Koss, 1984) and then were grouped in to broader regions (Figure 2) if the relationships between the sensor and FEM or FRM measurements were not significantly different. Uncorrected PurpleAir sensors in this work overestimate $PM_{2.5}$ across U.S. regions (MBE greater than 0 µg m$^{-3}$; Figure 6). Figure 6 shows the regional performance as displayed on PurpleAir.com ("raw"), with a linear correction, and with the final selected additive RH correction (Eq. 10). Linear regression improves the RMSE in each region and the MBE also decreases in all regions except for Alaska. When adding the RH term to the linear regression, the bias is further reduced across

all regions and the RMSE improves across all regions except for the Southeast where it increases slightly (<10%). Alaska shows the strongest underestimate, only 1 µg m⁻³ on average, which appears to be driven be strong underestimates of $PM_{2.5}$ (by >5 µg m⁻³) which occur in the winter between November and February with subfreezing temperatures (-1 to -25 °C). Plantower reports that the operating range of the sensors is -10 to 60 °C so this may contribute to the error (Plantower, 2016). However, other states see subfreezing temperatures (6% of U.S. dataset) but most of this subfreezing data from other states does not have a strong negative bias (>98%) even the points that are in a similar temperature range to the Alaska data. This could suggest unique winter particle properties or sensor performance in Fairbanks. However, information on particle size distribution or composition is not available.

To aid our air monitoring partner agencies, we have also provided state by state performance results in the supplement (Section S2 and Figure S10). It is important to note that the reported performance may not accurately summarize state-wide performance in states with less than a year of data or those with a limited number of collocation sites.

### 5.3.3    Error in corrected data by AQI category

We summarize the performance of the sensors across the U.S. using the U.S. daily AQI categories (Federal Register, 1999). For this analysis we use the data corrected using the LOSO withholding where a final correction is built for all but one state and then applied to the withheld state. This allows us to better understand how the correction will perform in locations not included in our analysis. Figure 7 shows the AQI as generated by the corrected PurpleAir (in colors) versus the AQI generated by the FEM or FRM with vertical lines indicating the break points between categories. With correction, the PurpleAir sensors report the correct AQI category 91% of the time while underestimating by one category 3% of the time and overestimating by one category 6% of the time. Many of these categorical disagreements occur near the AQI category break points where the estimates between the sensor and FEM or FRM measurements are within a few µg m⁻³ but this difference breaks the concentrations into different categories. In the moderate AQI category, as measured by the FEM or FRM, we see examples (in orange) where the PurpleAir shows large overestimates near the border between the good and moderate categories. These points represent 0.1% of the total dataset and are from sensors in Washington and California during times in both the summer (August) and winter (November-January). This overestimate suggests that the PurpleAir is measuring more light scattering per mass than is typical in other U.S. locations. Future work is needed to identify the factors affecting the strong sensor overestimates during these short time periods. From a public health perspective, however, there is more concern when the sensor strongly underestimates the $PM_{2.5}$ AQI.

There is also some underestimation in the moderate category. There are daily AQI values near the transition between moderate and unhealthy for sensitive groups where the PurpleAir is still in the good category (green). These occur primarily in the West (California). Past work has shown that PurpleAir sensors, and their internal Plantower PMS5003 sensors, underestimate $PM_{2.5}$ mass from larger particles including during dust impacted days (Kuula et al., 2020;Robinson, 2020;Kosmopoulos et al., 2020). Dust impacts may be driving the underestimates on these days in the West because it is harder for larger particles to be sampled by the low flowrate fans, especially under higher windspeeds, and also because larger particles

scatter less light per mass than smaller particles. Future work will be needed to develop an indicator and methodology to address the issue of dust. It may be impossible to use only the data from the PurpleAir (Duvall et al., 2020;Pawar and Sinha, 2020). Alternatively, regional information from satellites or other sources or more advanced sensor hardware may be able to improve these measurements in the future. In all, this represents <1% of the dataset. Typically, the sensors provide accurate estimates of the AQI category and have the potential to provide additional spatial density across the U.S. where regulatory and AirNow monitors are not currently found.

### 5.3.4  Comparison to existing correction equations

Lastly, we compared the U.S.-wide correction equation to others currently (as of March 11, 2021) available on the PurpleAir map and to recent smoke impacted corrections. The map currently defaults to displaying the raw [cf_atm] PM$_{2.5}$ data; however, a drop down also allows you to select from four corrections, the "US EPA" correction (detailed in this paper), the Lane Regional Air Protection Agency (LRAPA) correction or the AQ&U correction, both of which use this raw [cf_atm] data in their correction equations (Sayahi et al., 2019;Giles, 2020), and the woodsmoke correction (Robinson, 2020) that uses the cf_1 data. The U.S.-wide correction, presented here, and displayed on PurpleAir.com as the "US EPA" correction uses the [cf_1] data. The difference between these two data channels was discussed in Section 2.2.1 and Figure 3.

The LRAPA correction is a basic linear equation developed by the Lane Regional Air Protection Agency in Oregon while the PurpleAir sensor was being impacted by wood smoke from home heating in the winter. It was developed specifically for LRAPA's local airshed. The LRAPA correction is similar to our U.S.-wide correction equation without an RH term; PM$_{2.5}$ = 0.5*PA$_{cf\_atm}$ - 0.66 (LRAPA, 2018). Assuming an RH of 70%, both corrections would yield similar results until roughly 25 µg m$^{-3}$ when the [cf_atm] and [cf_1] start to disagree; however, in reality the relationships would vary as the RH varied. After this threshold, the LRAPA correction will result in lower concentrations which underestimate PM$_{2.5}$ as measured by the FRM or FEM in our dataset by about 33%. Applying this correction to our dataset results in an underestimate of PM$_{2.5}$ by 3 µg m$^{-3}$ (MBE= -3 µg m$^{-3}$, 34%) on average with more scatter as quantified by the RMSE (LRAPA= 4 µg m$^{-3}$, US correction=3 µg m$^{-3}$).

The AQ&U correction is a linear correction developed for Salt Lake City, UT (Sayahi et al., 2019). The AQ&U correction is updated as additional data becomes available and is, at the time of this article, PM$_{2.5}$=0.778*PA$_{cf\_atm}$+2.65 (Sayahi et al., 2019). At high concentration (>25 µg m$^{-3}$) the slope in the AQ&U and U.S.-wide corrections are fairly similar (i.e. [AQ&U] 0.778*PA$_{cf\_atm}$=[U.S.-wide equation] 0.52*PM$_{cf\_1}$ = 0.52*3/2*PM$_{cf\_atm}$ = 0.795*PA$_{cf\_atm}$); at lower concentrations the AQ&U correction may provide somewhat higher estimates, although, it will depend on the RH. Since RH is typically low in Salt Lake City this may lead to some of the overestimate in using this equation in more humid parts of the country. Applying this correction to our dataset results in an overestimate of PM$_{2.5}$ of 4 µg m$^{-3}$ (MBE= 4 µg m$^{-3}$, 51%) with more scatter as quantified by the RMSE (AQ&U=6 µg m$^{-3}$, U.S. correction=3 µg m$^{-3}$).

The woodsmoke correction is a linear correction developed for domestic wood-heating in New South Wales, Australia (Robinson, 2020). The equation is similar to that generated in this work with a slope that is 5% higher and a slightly lower intercept even considering the inclusion of an RH term in our equation (0.55*PM$_{cf\_1}$ + 0.53). Overall, applying this equation

to our dataset results in a slight underestimate of PM$_{2.5}$ by 1 µg m$^{-3}$ (MBE= -1 µg m$^{-3}$, 12%) on average with a similar scatter as measured by RMSE (both Woodsmoke and US correction =3 µg m$^{-3}$).

The U.S.-wide correction developed in this work will likely provide a more accurate correction across the U.S. in comparison to selecting either region-specific correction or the correction built for woodsmoke in Australia. The U.S. correction is more robust in part because the RH term can help account for meteorological variation across the U.S.

       Air sensors are potentially of the greatest use during wildland fire smoke impacted times (Holm et al., 2020;Durkin et al., 2020;Holder et al., 2020;Delp and Singer, 2020;Davison et al., 2021). A recent paper developed a smoke specific correction

(0.51*PA$_{cf\_1}$ - 3.21) for PM$_{2.5}$ concentrations from PurpleAir sensors based on smoke impacts from multiple types of fires in the U.S. (Holder et al., 2020). The slope is within 3% of that calculated for the U.S.-wide correction. In the smoke study, RH was found not to significantly improve the model. This lack of significance is likely because the data did not come from as diverse of locations and seasons as the U.S.-wide dataset. The median RH in Holder et al. was around 40% which would make the U.S. correction intercept +2.312. The intercepts differ by 5 µg m$^{-3}$. Since the U.S. correction was built on more low

concentration data, it likely provides a better constrained estimate of intercept and this difference will be a small percent difference under high concentration smoke events. At a PurpleAir PM$_{2.5}$ concentration of 300 µg m$^{-3}$, the smoke correction would give an estimate of 150 µg m$^{-3}$ while the U.S.-wide correction would give an estimate of 160 µg m$^{-3}$, a difference of only 6%. Another recent paper developed smoke adjustment factors, linear adjustments with zero intercepts, for a variety of fires in California and Utah ranging between 0.44 and 0.53 (Delp and Singer, 2020). The slope calculated in our study is also

within this range. Although there was limited smoke data included in the analysis in this paper, the similarity between the correction generated here and under smoke impacted times suggests that this equation will work well under smoke conditions.

**5.4     Limitations and implications**

       Because PM sensors use an optically based detector, they will never be able to perfectly capture the PM$_{2.5}$ mass because of the many factors affecting the optical-mass relationship (Liu et al., 2008). However, there are a number of higher complexity

optical methods that are used frequently with adequate accuracy (Heintzenberg et al., 2006;Chung et al., 2001). Nephelometers are used for routine monitoring in some parts of the U.S. (OR DEQ, 2020) and are frequently used in health effects research (Delfino et al., 2004). The Teledyne T640 and T640x and Grimm EDM 180 are optically based monitors have been approved as FEMs in the past decade (US EPA, 2016). Humidity tends to induce large errors in these types of measurements (Chakrabarti et al., 2004;Day and Malm, 2001) which is addressed using a dryer or humidity control in FEMs (US EPA, 2016). The

PurpleAir sensor provides minimal humidity control due to the higher internal temperature caused by the small volume containing the electronics.

       The only reason a single U.S. correction is possible is because the dual Plantower sensors within the PurpleAir sensor typically have strong precision. It would not be possible to develop a single correction for sensors with high error or more variability among identical units. In addition, having two Plantower sensors in each PurpleAir sensor enables a data cleaning

step based on sensor health, where we compare the A and B channels and exclude data where they agree poorly (Section 3.2.3). Alternative approaches would be necessary for devices with only a single PM sensor. A similar approach, as outlined in this

work, could be applied to develop U.S.-wide corrections for other sensors with collocation data from across the U.S. However, similar or better precision among identical units and quality assurance methods that check sensor health and flag questionable data would be needed. Adding data from additional types of air sensors could further increase the spatial knowledge of air quality across the U.S. moving forward.

The proposed PurpleAir correction in this work relies on RH data and in some cases the internal RH sensor may drift or fail. Users have two options if no valid RH data is reported: 1) discard data when the RH is missing or 2) to assume a RH based on typical ambient conditions in the U.S. or specific geographical area. In our dataset, <1 % of the RH data was missing but this may happen more often for individual sensors or over time as RH sensors fail. There will be additional uncertainty in the measurement if the RH term is not available but substituting a value of 50% may limit this error. RH sensor drift should result in less error than full RH sensor failure and future work should further explore the performance of the RH sensor.

Although this dataset includes sites throughout the U.S. (see Figure 2), some regions are oversampled while others are undersampled. The oversampled areas include Iowa and California (especially the South Coast Air Basin) which together represent 58% of the dataset. Both Iowa and the South Coast Air Basin have a higher fraction of nitrate in $PM_{2.5}$ than many other areas of the U.S., which may impact the hygroscopicity of particles represented in this dataset. Utah in winter has a similar composition which may be why the AQ&U correction is comparable. Undersampled areas as defined by the NOAA climate regions include southern parts of the South (i.e. Texas, Louisiana, Mississippi), the Northern Rockies (i.e. North and South Dakota, Nebraska and Wyoming), and also the Ohio Valley (i.e. Missouri, Illinois, Indiana, Ohio, Kentucky, and Tenessee) where $PM_{2.5}$ may have different optical properties due to different air pollution emission sources. In addition, only three sites in the dataset are classified as rural sites. It may be beneficial to collocate additional sensors in rural areas especially as sensors may provide the most value where government monitors are sparse. Furthermore, other localized source-oriented locations such as near major roadways, airports, and ports are not well-represented in this dataset and may not be well characterized by our correction. The Alaska site is one location included in this work where additional collocated sensors, along with additional information about particle properties, could help to better understand whether the proposed correction can be improved. Future work may be able to develop additional correction factors based on aerosol types through a concerted effort to collocate sensors with the Chemical Speciation Network (CSN) or Interagency Monitoring of Protected Visual Environments (IMPROVE) network. The applicability of this correction to areas outside of the U.S. is also uncertain because much higher concentrations of $PM_{2.5}$ (likely with different size distributions and chemical components) are common throughout the globe (van Donkelaar et al., 2016). In addition, there is uncertainty in how higher concentrations may damage sensors or lead to faster sensor aging, potentially requiring more regular maintenance and/or replacement (Wang et al., 2020a).

Since PurpleAir sensors were operated by air monitoring agencies, the dataset used for this work is an ideal dataset with potentially better performance than PurpleAir sensors operated by the general public. Every sensor location was confirmed, unlike sensors on the PurpleAir map that may have been relocated, moved indoors, or assigned an incorrect location for privacy reasons. In addition, air monitoring agencies have taken care to appropriately site the PurpleAir sensors in places with good air flow which may not be the case for all community deployed sensors. Future work may be needed to explore how

to identify and flag sensors with incorrect locations and poor siting. In some cases, the performance of the PurpleAir sensors used in this project was evaluated before deployment to check for any issues between the A and B channels when the sensors arrived from PurpleAir. In many cases, the agencies hosting the PurpleAir sensors check the data regularly and may immediately address performance issues. This may result in a higher data completeness and better performance between the A and B channels than would be seen by sensors operated by the general public; however, our AB comparison methodology should flag these performance issues. The criteria for this work were specifically stringent so that the model would be built on reliable data. Future work could explore loosening the criteria for AB agreement and data completeness.

During regulatory monitoring, the site operator plays a significant role in annotating the site data, metadata, and in maintaining records that document the monitoring effort. Although we received some of these notes from agencies operating sensors for this project, we would not expect any of this data to be present for publicly available sensors on the PurpleAir map. Since the insights of the site operator are not incorporated into the PurpleAir data from Thingspeak, the job of annotating the raw data record passes to the data analyst, someone with likely little on the ground knowledge of how the sensor is being operated. As a result, some questions that arise that could explain sensor performance will be impossible to answer. Although some automated checks like the A and B channel comparison can be applied, we will not be able to attain the same level of confidence as a monitor with a site operator documenting and quality assuring data.

There are still unknowns about sensor performance over the long-term and during extreme events. Large performance deteriorations were not seen in this dataset with sensors up to two years old, but more targeted analysis should be completed especially as the network continues to age. This work was conducted using 24-hour averages. It can be more challenging to develop accurate corrections using shorter time averaged data (e.g. 1-hour or 2-minute averages) due to limitations in FRM measurements and increased noise in higher time resolution FEM measurements. Additional work is currently being done to understand the performance of this correction when applied to shorter time averaging intervals and during high concentration smoke impacted events when public interest in air quality is high and health/environmental impacts may be of concern.

This correction equation is currently being applied to the low-cost sensor data (currently supplied by PurpleAir) on the AirNow Fire and Smoke Map (fire.airnow.gov), along with similar data cleaning methods, and data is presented in the form of the NowCast AQI. This allows the public to see greater spatial variability in $PM_{2.5}$ AQI than would be available with only AirNow monitors. The AirNow Fire and Smoke Map will be updated based on user feedback and as additional data become available to improve the correction and data cleaning methods. This website was well received by state, local, and tribal partner air monitoring agencies and the public, and received over 7 million page views in the first three months. A current screenshot in available in the supplement (Figure S11).

## 6    Conclusions

This work developed an effective methodology for cleaning $PM_{2.5}$ data from the PurpleAir sensor by removing poorly performing sensors and short-term outlier concentration measurements using channel A and B comparisons. A single U.S. correction model for the PurpleAir sensor was developed which includes additive correction terms using [cf_1] $PM_{2.5}$ and on-

board RH data. The U.S. correction improves PurpleAir measurement performance, reducing the 24-hour averaged PM$_{2.5}$ data RMSE from 8 to 3 µg m$^{-3}$ when evaluated against regulatory measurements across the U.S. and reduced the bias to $\pm 3$ µg m$^{-3}$ when validated on a state-by-state basis and to $\pm 1$ µg m$^{-3}$ when evaluating by region. With correction, the PurpleAir reports the 24-hour averaged PM$_{2.5}$ AQI within 1 category 100% of the time and reports the correct category 91% of the time. Although no previous work had attempted a broadly applicable correction, the correction developed in this paper is similar to those developed for specific locations or sources (i.e. smoke) strengthening the confidence that this correction is applicable across the U.S. This national evaluation suggested that any corrections that are not strictly local likely need to include RH or other environmental factors to represent the wide range of conditions that can occur in the U.S. Corrected PM$_{2.5}$ data from the PurpleAir sensor can provide more confidence in measurements of ambient PM$_{2.5}$ concentrations for a wide range of potential applications including exposure assessments and real-time public health messaging. PurpleAir PM$_{2.5}$ data with this U.S.-wide correction is currently displayed on a pilot sensor data layer on the AirNow Fire and Smoke Map (fire.airnow.gov).

More work is needed to understand if similar corrections can be developed for other sensor types. If other highly precise sensors with duplicate measurements are identified, similar methodology could be used to develop data cleaning steps and a nationwide correction. However, it is recommended that sensors are first collocated with reference measurements across the U.S. (i.e. FEM and FRM methods), ideally for a year or more to reduce uncertainties caused by seasonal influences, over a range of meteorological conditions, and across PM concentration ranges and source types. Most other sensor types do not contain duplicate PM$_{2.5}$ measurements which will make ensuring their data quality more challenging and more complex methods of data cleaning may be required or similar data quality may not be possible. Developing correction methods and data cleaning methodology for additional sensor types could further increase the amount of data available to communities, epidemiologists, decision makers, and others.

## 7    Data availability

Data is available at https://catalog.data.gov/dataset/epa-sciencehub (Barkjohn, 2021).

## 8    Author contribution

KB and AC conceptualized the work. KB and BG curated the data. KB completed the formal analysis, developed the methods and figure visualizations. AC acquired funding, cultivated relationships, and launched the field sampling campaign. KB, AC, BG wrote the original draft, reviewed, and edited.

## 9    Competing interests

The authors declare that they have no conflict of interest.

## 10    Acknowledgments

This work would not have been possible without the partnership of many SLT air monitoring agencies and other partners, including: State of Alaska, Citizens for Clean Air (Alaska), Maricopa County Air Quality Department, San Luis Obispo County

Air Pollution Control District, Mojave Desert Air Quality Management District, California Air Resources Board, Santa Barbara County Air Pollution Control District, Ventura County Air Pollution Control District, Colorado Department of Public Health and Environment, Delaware Division of Air Quality, Sarasota County Government, Georgia Environmental Protection Division, Iowa Department of Natural Resources, Polk and Linn County (Iowa) Local Programs, the State Hygienic Laboratory at the University of Iowa, Kansas Department of Health & Environment, Missoula County, Montana Department of Environmental Quality, Forsyth County Office of Environmental Assistance & Protection, Clean Air Carolina, Quapaw Nation, Oklahoma Department of Environmental Quality, Virginia Department of Environmental Quality, State of Vermont, Puget Sound Clean Air Agency, Wisconsin Department of Natural Resources. These agencies and organizations provided data and shared their experiences in using the PurpleAir sensors in their jurisdictions. We would also like to thank Sean Fitzsimmons at Iowa Air Quality Bureau, Ian VonWald who is an ORISE postdoc hosted by EPA, Samuel Frederick who is an ORAU student services contractor to EPA, and Amara Holder, Rachelle Duvall, and Gayle Hagler at EPA for their help. We thank PurpleAir for maintaining and managing the repository of public and private data. We thank Adrian Dybwad and the PurpleAir staff for their time discussing this work and the addition of this correction as an option on the PurpleAir website. We thank the United States Forest Service and EPA's AirNow team for the incorporation of the air sensor pilot data layer on the AirNow Fire and Smoke map. We are grateful to our referees for their constructive input.

This project was supported in part by an appointment to the Research Participation Program at the EPA ORD's Center for Environmental Measurements and Modeling (CEMM), administered by the Oak Ridge Institute for Science and Education through an interagency agreement between the U.S. Department of Energy and EPA.

## 11 Disclaimer

The views expressed in this paper are those of the author(s) and do not necessarily represent the views or policies of the U.S. Environmental Protection Agency. Any mention of trade names, products, or services does not imply an endorsement by the U.S. Government or the U.S. Environmental Protection Agency. The EPA does not endorse any commercial products, services, or enterprises.

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

Table 1. Summary of the dataset used to generate the U.S-wide PurpleAir correction equation after 3 sensors with large A, B channel discrepancies were removed. PM$_{2.5}$ concentrations from both the FEM or FRM and the raw PurpleAir (PA), temperature (T) and relative humidity (RH) are summarized as median (min, max).

| State | Start Date | End Date | # of PA | # of Sites | # of points | FEM or FRM | FEM or FRM PM$_{2.5}$ (µg m$^{-3}$) | PA PM$_{2.5}$ (µg m$^{-3}$) | PA T (°C) | PA RH (%) |
|---|---|---|---|---|---|---|---|---|---|---|
| CA | 11/29/2017 | 12/29/2019 | 13 | 12 | 3762 | Both | 6 (-2,109) | 7 (0,250) | 22 (6,42) | 45 (2,100) |
| IA | 9/29/2017 | 1/13/2020 | 9 | 5 | 3762 | Both | 10 (0,36) | 19 (0,69) | 11 (-27,35) | 55 (21,100) |
| WA | 10/16/2017 | 10/28/2019 | 3 | 3 | 1035 | FEM | 6 (0,41) | 8 (0,89) | 13 (-2,30) | 63 (26,84) |
| AZ | 11/9/2018 | 12/31/2019 | 3 | 3 | 895 | Both | 7 (1,43) | 6 (0,74) | 24 (9,44) | 26 (5,73) |
| WI | 1/1/2019 | 11/18/2019 | 6 | 4 | 811 | Both | 6 (1,32) | 9 (1,64) | 18 (-25,33) | 53 (31,82) |
| NC | 3/25/2018 | 10/24/2019 | 1 | 1 | 700 | Both | 7 (0,20) | 13 (1,43) | 25 (-1,35) | 48 (16,79) |
| AK | 11/7/2018 | 9/30/2019 | 3 | 1 | 369 | FRM | 4 (0,60) | 4 (0,131) | 8 (-25,29) | 47 (21,76) |
| KS | 3/13/2019 | 9/30/2019 | 3 | 1 | 306 | FEM | 9 (2,33) | 11 (0,50) | 24 (9,34) | 52 (30,71) |
| DE | 7/27/2019 | 11/18/2019 | 1 | 1 | 205 | Both | 7 (1,17) | 9 (1,35) | 25 (6,35) | 51 (34,75) |
| OK | 7/10/2019 | 11/18/2019 | 2 | 2 | 190 | Both | 9 (1,25) | 11 (1,35) | 30 (1,38) | 57 (29,86) |
| GA | 8/2/2019 | 11/18/2019 | 1 | 1 | 184 | Both | 9 (3,18) | 15 (5,34) | 29 (5,36) | 55 (44,77) |
| VT | 3/30/2019 | 9/30/2019 | 1 | 1 | 146 | Both | 6 (2,18) | 8 (1,31) | 24 (12,34) | 52 (36,71) |
| FL | 5/31/2019 | 9/30/2019 | 1 | 1 | 119 | FEM | 6 (3,17) | 5 (1,25) | 32 (29,35) | 60 (49,73) |
| CO | 8/22/2019 | 11/18/2019 | 1 | 1 | 113 | both | 7 (2,25) | 6 (1,45) | 18 (-5,32) | 33 (18,70) |
| VA | 10/27/2019 | 12/29/2019 | 1 | 1 | 30 | FRM | 5 (2,20) | 10 (2,41) | 12 (8,25) | 48 (35,65) |
| MT | 12/3/2019 | 12/10/2019 | 1 | 1 | 8 | FEM | 10 (5,15) | 22 (6,36) | 4 (2,6) | 54 (42,62) |
| All | 9/29/2017 | 1/13/2020 | 50 | 39 | 12635 | both | 7 (-2,109) | 10 (0,250) | 19 (-27,44) | 51 (2,100) |

Table 2. Correction equation forms considered and the root mean squared error (RMSE). The best performing model from each increasing complexity (as indicated with *) was validated using withholding in the next sections (Section 5).

| Name | Equation | RMSE ($\mu g\ m^{-3}$) (cf_1) | RMSE ($\mu g\ m^{-3}$) (cf_atm) |
|---|---|---|---|
| linear | $PA = PM_{2.5}*s_1 + b$ | 2.88* | 3.01 |
| +RH | $PA = s_1*PM_{2.5} + s_2*RH + i$ | 2.52* | 2.59 |
| +T | $PA = s_1*PM_{2.5} + s_2*T + i$ | 2.84 | 2.96 |
| +D | $PA = s_1*PM_{2.5} + s_2*D + i$ | 2.86 | 2.99 |
| +RH*T | $PA = s_1*PM_{2.5} + s_2*RH + s_3*T + s_4*RH*T + i$ | 2.52 | 2.60 |
| +RH*D | $PA = s_1*PM_{2.5} + s_2*RH + s_3*D + s_4*RH*D + i$ | 2.52 | 2.60 |
| +D*T | $PA = s_1*PM_{2.5} + s_2*D + s_3*T + s_4*D*T + i$ | 2.51* | 2.61 |
| +RH*T*D | $PA = s_1*PM_{2.5} + s_2*RH + s_3*T + s_4*D + s_5*RH*T + s_6*RH*D + s_7*T*D + s_8*RH*T*D + i$ | 2.48* | 2.57 |
| PM*RH | $PA = s_1*PM_{2.5} + s_2*RH + s_3*RH*PM_{2.5} + i$ | 2.48* | 2.53 |
| PM*T | $PA = s_1*PM_{2.5} + s_2*T + s_3*T*PM_{2.5} + i$ | 2.84 | 2.96 |
| PM*D | $PA = s_1*PM_{2.5} + s_2*D + s_3*D*PM_{2.5} + i$ | 2.86 | 3.00 |
| PM* Nonlinear RH | $PA = s_1*PM_{2.5} + s_2\frac{RH^2}{(1-RH)}*PM_{2.5} + s_3*\frac{RH^2}{(1-RH)} + i$ | 2.86 | 2.99 |
| PM*RH*T | $PA = s_1*PM_{2.5} + s_2*RH + s_3*T + s_4*PM_{2.5}*RH + s_5*PM_{2.5}*T + s_6*RH*T + s_7*PM_{2.5}*RH*T + i$ | 2.46* | 2.53 |
| PM*RH*D | $PA = s_1*PM_{2.5} + s_2*RH + s_3*D + s_4*PM_{2.5}*RH + s_5*PM_{2.5}*D + s_6*RH*D + s_7*PM_{2.5}*RH*D + i$ | 2.54 | 2.57 |
| PM*T*D | $PA = s_1*PM_{2.5} + s_2*T + s_3*D + s_4*PM_{2.5}*T + s_5*PM_{2.5}*D + s_6*T*D + s_7*PM_{2.5}*T*D + i$ | 2.52 | 2.63 |
| PM*RH*T*D | $PA = s_1*PM_{2.5} + s_2*RH + s_3*T + s_4*D + s_5*PM_{2.5}*RH + s_6*PM_{2.5}*T + s_7*T*RH + s_8*PM_{2.5}*D + s_9*D*RH + s_{10}*D*T + s_{11}*PM_{2.5}*RH*T + s_{12}*PM_{2.5}*RH*D + s_{13}*PM_{2.5}*D*T + s_{14}*D*RH*T + s_{15}*PM_{2.5}*RH*T*D\ i$ | 2.42* | 2.51 |

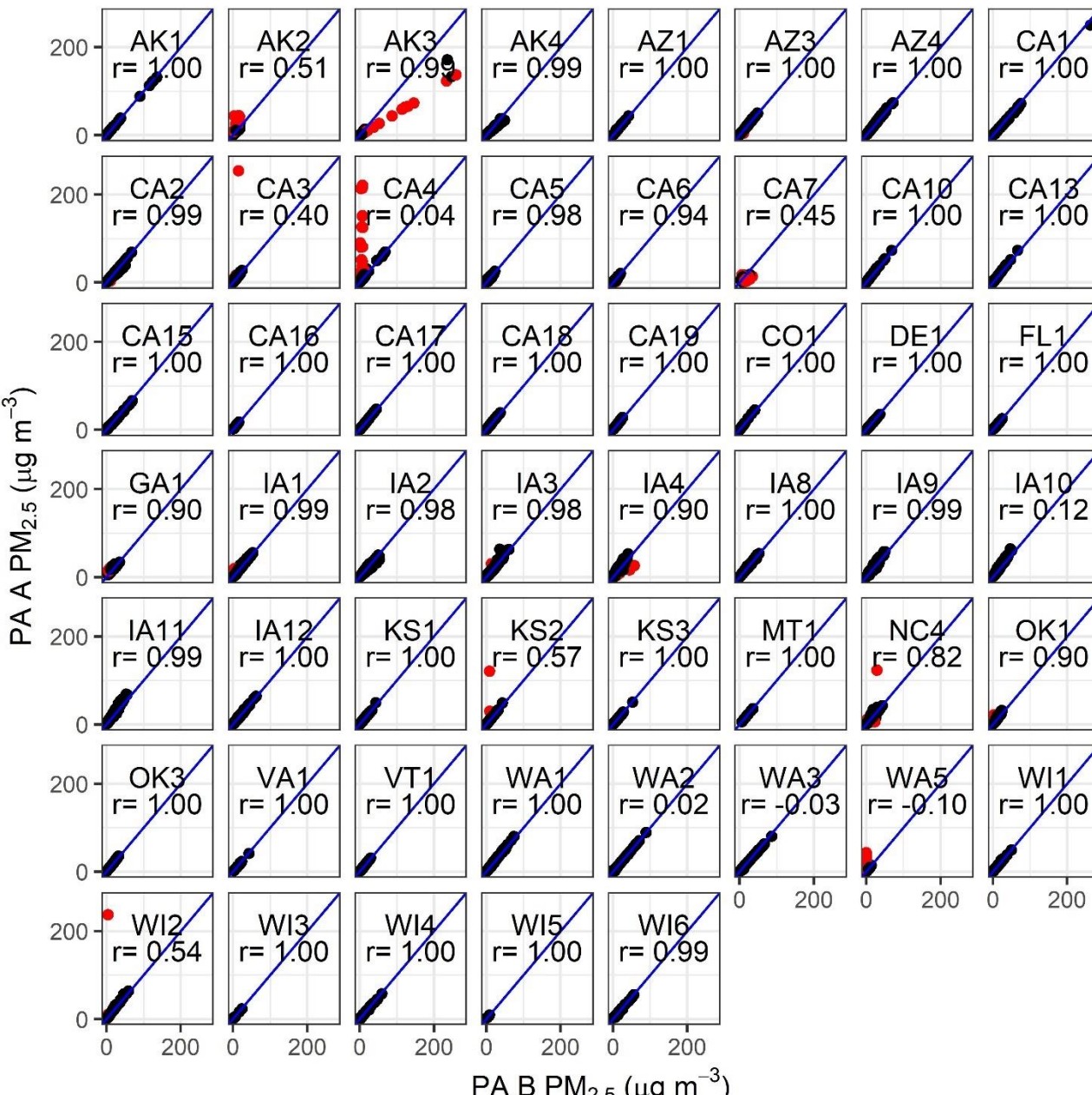

Figure 1. Comparison of 24-hour averaged PM$_{2.5}$ data from the PurpleAir A and B channels. Excluded data (2.1%) are shown in red and represent data points where channels differed by more than 5 µg m$^{-3}$ and 61%. AK3, CA7, WA5 were excluded from further analysis. Pearson correlation (r) is shown on each plot.

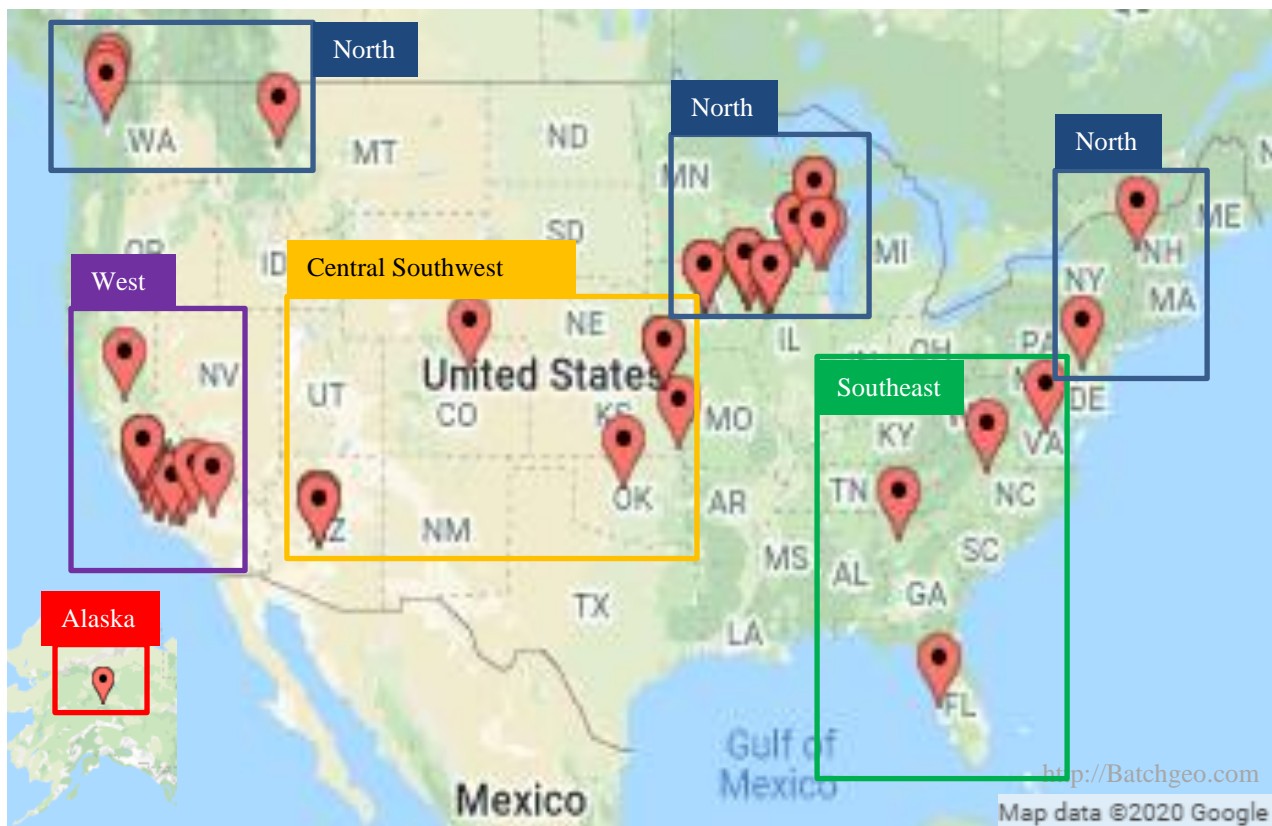

**Figure 2.** State, local, and tribal (SLT) air monitoring sites with collocated PurpleAir sensors. Includes regions used for correction model evaluation.

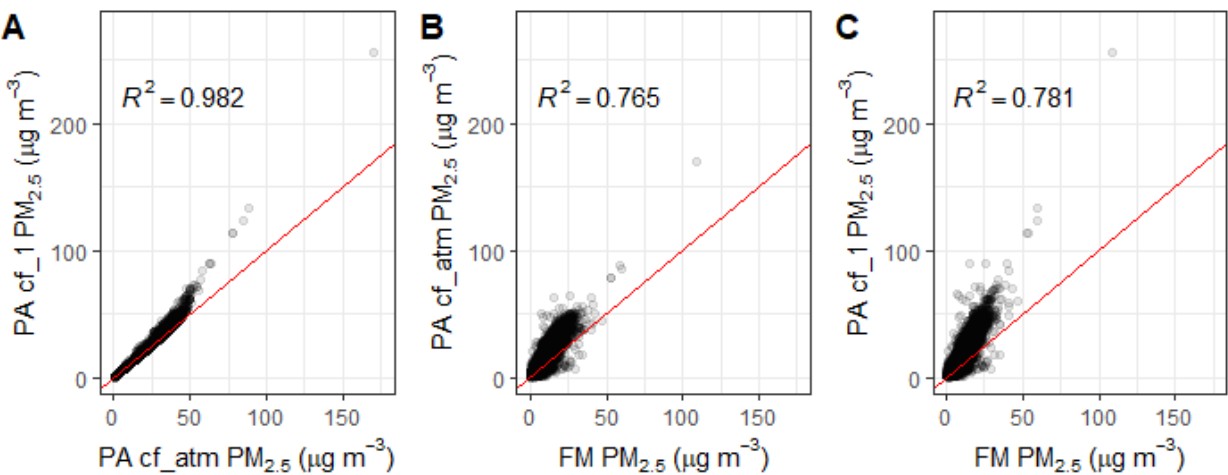

**Figure 3.** Comparison of the 24-hour raw PurpleAir (PA) cf_1 and cf_atm PM$_{2.5}$ outputs (A) and both outputs compared to
the FEM or FRM PM$_{2.5}$ measurements (B and C) across all sites with the 1:1 line in red.

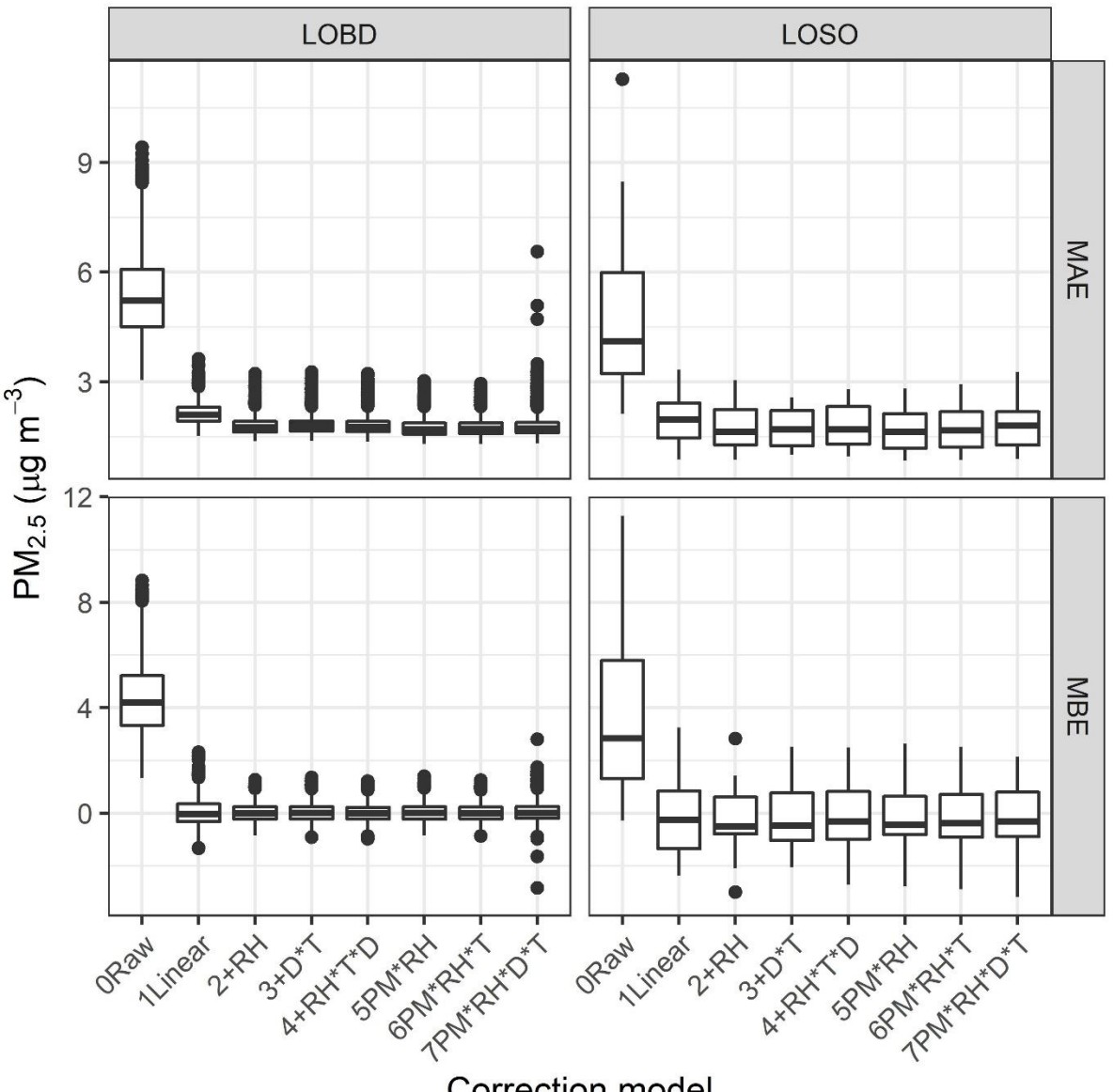

**Figure 4.** Performance statistics including mean bias error (MBE) and mean absolute error (MAE) are shown by correction method (0-7), where each point in the boxplot is the performance for either a 12-week period excluded from correction building ("LOBD"), or a single state excluded from correction building ("LOSO").

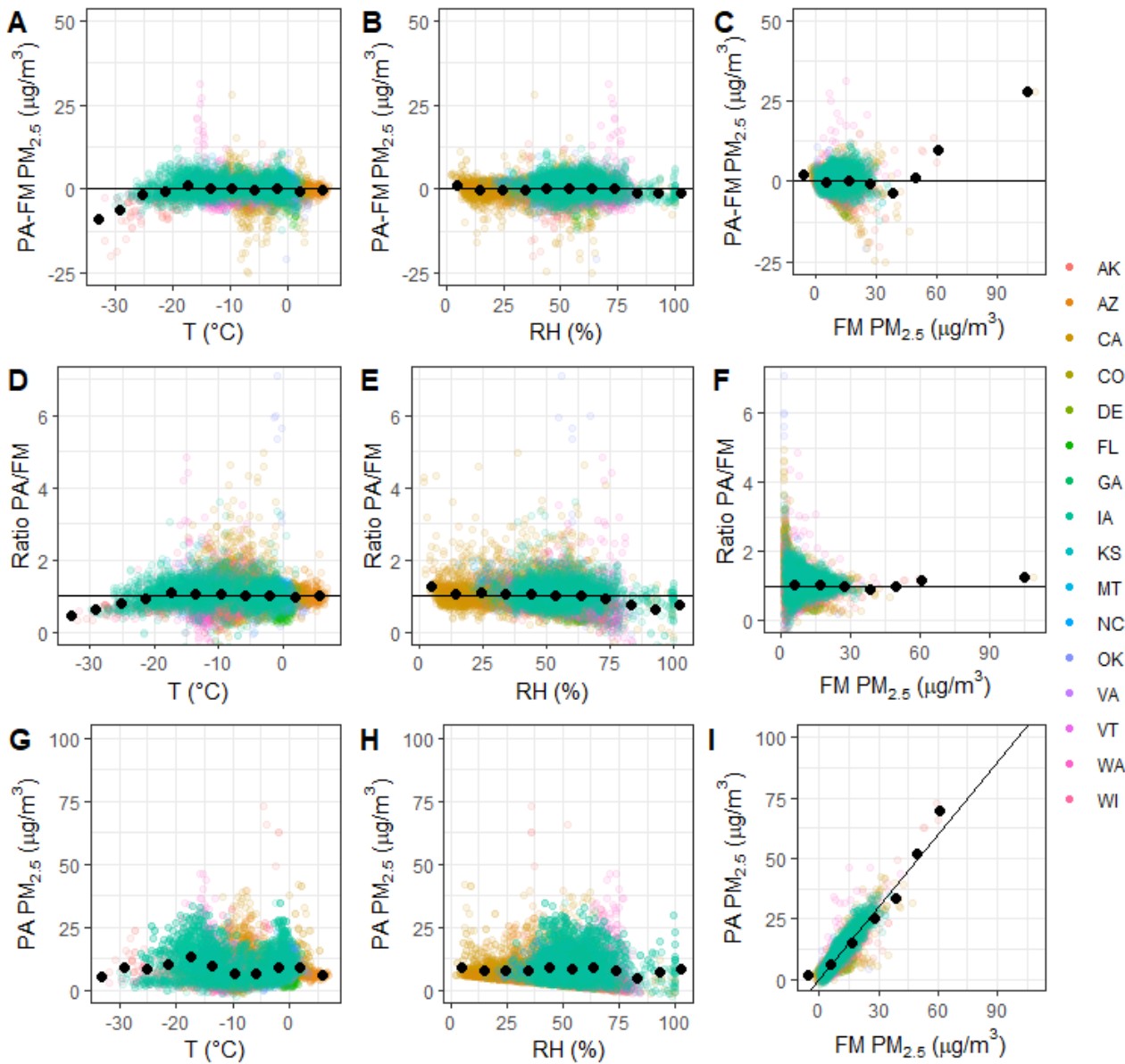

**Figure 5.** Error and ratio between corrected PurpleAir (PA) and FRM or FEM measurements are shown along with corrected PurpleAir PM$_{2.5}$ data (corrected using Eq. 10) as influenced by temperature, RH, and FRM or FEM PM$_{2.5}$ concentration. Colors indicate states, and black points indicate averages in 10 bins.

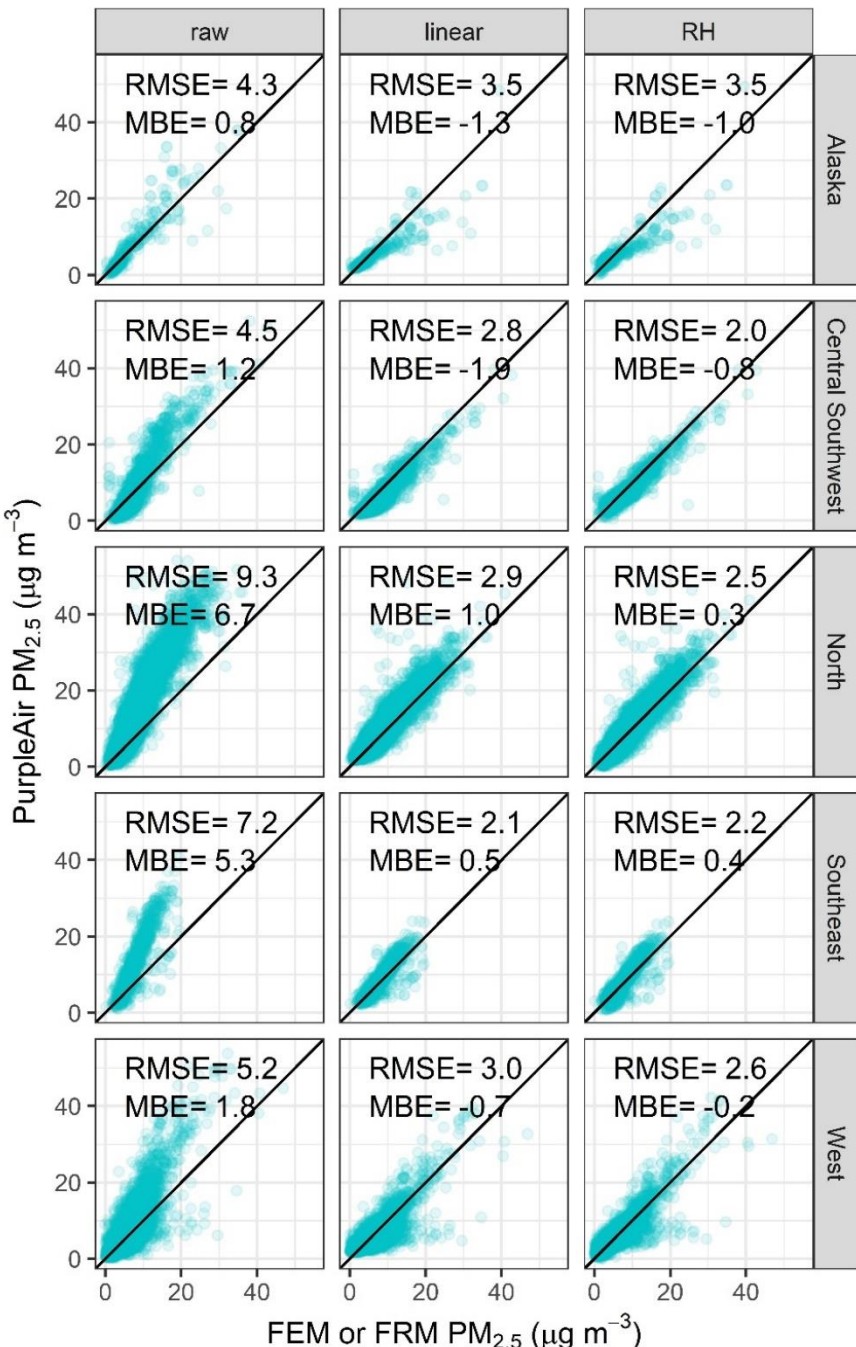

Figure 6. Scatterplot of the daily FEM or FRM PM$_{2.5}$ data with the PurpleAir data by U.S. region (see Figure 2) prior to any correction, after applying a linear correction, and after applying the final correction including RH. Data were corrected using the models built for the full dataset.

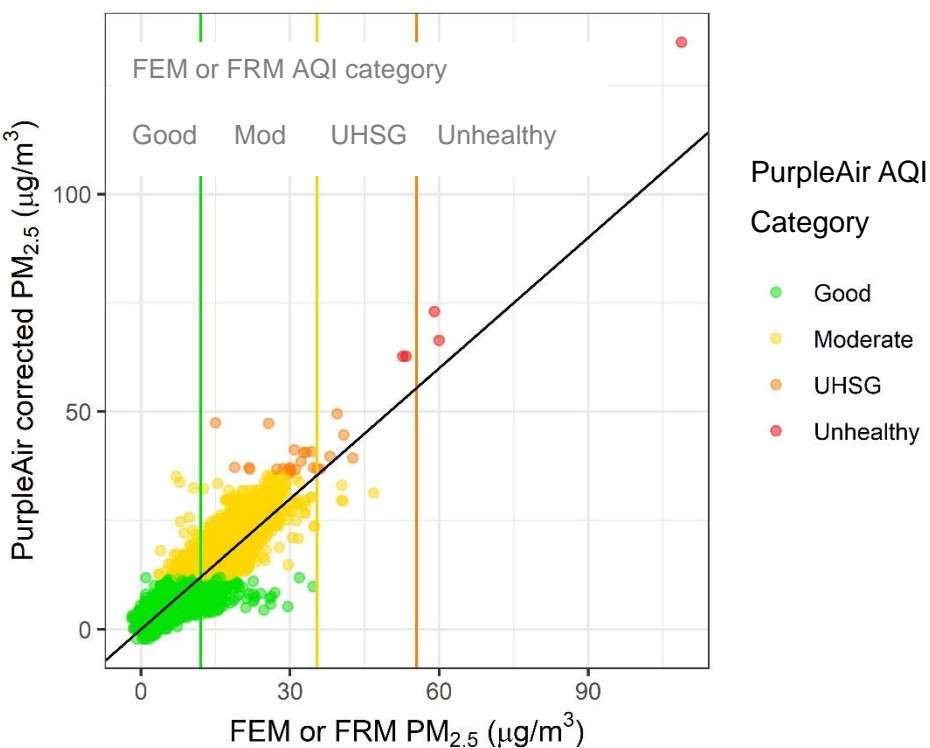

Figure 7. 24-hr AQI categories as measured by the corrected PurpleAir and the FEM or FRM for the full dataset generated
with the models built using LOSO withholding.