# Peer review of "Development and Application of a United States wide correction for PM2.5 data collected with the PurpleAir sensor"

_Atmospheric Measurement Techniques, 2020_

## Referee Comment (RC1) · Anonymous Referee #3 · 9 Dec 2020

The manuscript presents a development of a generalized correction equation for the PurpleAir PM2.5 sensors in the USA. The used data set includes 39 different sites in 16 states and 50 sensor units altogether. The number of recorded datapoints was 12,635 (24h-average, from 8 to 3762 per state). Only sites which had reference level data available no further than 50m away were included in the analysis.

Major comment:

The chosen analysis approach does not utilize the data as well as it could, and there is a strong case to be made that the generalized correction equation is not representative: in fact, a single generalized and true correction equation is most probably

impossible to form due to the site-specific differences in aerosol composition. Is it not obvious that two different sites entailing two completely different environments (e.g. rural background vs. urban city center) are not representative of each other? To add, the analysis does not consider possible seasonality, which may have a significant impact on aerosol composition: the shortest sensor data is composed of only 8 data points (MT, Table 1). When these issues are coupled with the notion that not all the factors affecting the sensor behavior are fully understood (line 296), generalizations should be made with extreme caution. In my opinion, the only way to address this issue is to generate correction equations for a few, most generalizable environments (e.g. urban, urban background, detached housing area, regional background) with known aerosol sources. This would also lend for a further investigation regarding the underlying reasons affecting sensor responses.

Technical comment:

In multiple linear regression independent variables should be independent. In my opinion, the fact that previous studies have misused linear regression does not warrant for a new research to continue misusing it. It reinforces the bad habit and undermines the quality and significance of the whole research line of low-cost aerosol sensors. Besides the relative humidity and temperature analysis, this applies also for the analysis regarding the binned particle counts (line 270).

Another misuse particularly characteristic for low-cost sensor studies is the use of $R^2$ as goodness-of-fit indicator in nonlinear regression. $R^2$ is not valid for nonlinear regression (line 295).

Recommendation: reject

---

## Referee Comment (RC2) · Anonymous Referee #1 · 27 Dec 2020

Review for Development and Application of a United States wide correction for PM2.5 data collected with the PurpleAir sensor

The paper Development and Application of a United States wide correction for PM2.5 data collected with the PurpleAir sensor by Barkjohn et al. is an evaluation of the PA units using reference sensors for PM2.5 values. Overall, the paper is well written and clear, I have some concerns and comments for things that hopefully you will be able to address and clarify

**General comments**

My biggest concern is about the fact you used the T and RH from the PA itself to perform your analysis. Even the company itself does not recommend using these values for ambient conditions for the same reason you mention in your paper. Also, you cite Holder et al. 2020 but even they state that "the PA temperature and RH measurement are interpreted as an internal rather than ambient measurement". You also enhance this comment in line 375.

Information on the location of co-located units (lat/long) will help the community to understand which purple air sensor you used (pubic), also if you could provide their name that will be important, at least for the public one.

Why 50 meters for distance for collocated units, 1 km is not good for that, other works used a distance larger than 50 m for collocated units. I am wondering how many more collocated units you could have gotten if you had a larger distance between units.

In table 1 you used 50 PA units but in Fig 1 you show more than 50, it is confusing. Maybe have in table 1 the full number of units

Why did you remove data from a station like Iowa, why not using as much data as you have as you could have a full range of T ad RH conditions?

PA company record on using the cf_atm so why also testing the other type?

In Section 3.5 you mention that the use of T for correction should be used on a local basis, but don't you think this is an important factor. In your work, you covered locations with wide T

conditions from cold Alaska to warm Florida is it possible that only changes in RH can represent the way to correct the PA data across the US? Also, from Table S2, it seems most of your RH were <100, only 3 PA units reach RH of 100 is it possible the lack of high RH harm your analysis.

Would be nice to see the distribution of RH and T from all dataset (not just for Iowa)

**Comments of figures and tables**

Many of your figures are pixelated

**Table 1 -** I assume PA data on table 1 is uncorrected. What do you mean by the wide PurpleAir correction equation?

**Fig 1 -** The quality of the fig itself is not great it is very pixelated, also R2 values on each plot would have helped to see the comparison between A and B. you should more h=than 50

**Fig 7** - Does the data represent the entire data set or one location?

**Specific comments**

Line 56-60: it seems you have too many references can you reduce the amount or at least separate them into multiple sentences

Line 85: you state that the data was until 2018 but some of your data was for 2019

Lines 110-114: the entire section is unclear, I do not understand what you mean, can you rewrite this part to make it clearer

 Line 235: what do you mean - *one ran multiple sensors in series*

---

## Referee Comment (RC3) · Anonymous Referee #2 · 27 Dec 2020

**GENERAL COMMENTS**

The authors present carefully-considered, practical guidance for performing quality assurance checks on PurpleAir data. The authors' model for correcting PurpleAir PM$_{2.5}$ data benefits from a large training dataset that spans many months and locations (39 sites in 16 U.S. states). Together, the quality assurance guidelines and correction model presented here can help users improve the accuracy of PM$_{2.5}$ data reported by PurpleAirs. There will likely be applications in which this U.S.-wide correction still does not produce sufficiently-accurate estimates of local ambient PM$_{2.5}$ concentrations, but I am not here to argue for letting the perfect be the enemy of the good.

**SPECIFIC COMMENTS**

1. Lines 79–82: I found these two sentences confusing. Are the PurpleAirs that are part of the "larger project" counted as PurpleAirs sent out by EPA (group 1 on line 77) or PurpleAirs independently operated by air monitoring agencies (group 2 on line 77)? Can the authors clarify how these PurpleAirs fit in? Alternatively, can the authors remove these sentences and simply describe the study as involving PurpleAirs "sent out by EPA to capture a wide range of regions and meteorological conditions" and "other collocated sensors"?

2. Lines 91–92: The authors state here that their analysis included data from 50 PurpleAirs at 39 sites in 16 states. On lines 87–88, the authors state that their search identified 42 sites in 14 states. Does that mean the 50 PurpleAirs/39 sites/16 states includes both PurpleAirs sent out by EPA and PurpleAirs identified in the search?

3. Lines 121–127: Was the "within 5 $\mu$g m$^{-3}$ or 61%" check performed on the 2-minute/80-second averages or on the 24-hour averages? Was this check performed on the cf_1 data or on the cf_atm data? Please clarify.

4. Section 2: Can the authors include some information on the breakdown of regulatory monitors and reference PM$_{2.5}$ measurements by type (FRM vs. FEM and type of FEM)? Maybe this information could be included in Tables S1–S2 and/or summarized in the text? FEM measurements do not always agree with FRM measurements and some FEMs might be more likely to deviate from FRMs than others. I downloaded the daily PM$_{2.5}$ data from the sites listed in Table S1, over the date ranges listed in Table S2, from the AQS. It looks like there were 17 FRMs, 16 BAMs, 16 spectrometers (T640s), and 6 TEOMs across the 39 sites. It looks like the breakdown of 24-hour measurements was approximately 10% FRM, 50% BAM, 30% spectrometer, and 10% TEOM.

5. Lines 153–154: It looks like there were 13 sites with both FRM and FEM data. It looks like there were three sites (53-033-0057, 53-061-1007, and 55-087-0009) with two FEM monitors and one site (19-113-0040) with two FEMs plus an FRM. What did

the authors do when multiple reference measurements (FRM or FEM) were available for a single 24-hour period at a single site? Were all 24-hour average measurements included as separate data points? Or was just one selected and, if so, which one?

6. Lines 154–155 and 470–474: Did the authors look at the agreement between 24-hour average FRM and FEM measurements for the dates and sites where both were available? It might be useful for comparison to the agreement between PA and FM data. For the sites and dates listed in Tables S1–S2, it looks like the RMSE for 24-hour FEM measurements, compared to FRM measurements, was 1.7 $\mu$g m$^{-3}$ for BAMs (n = 4134), 2.1 $\mu$g m$^{-3}$ for spectrometers (n = 4405), and 1.1 $\mu$g m$^{-3}$ for TEOMs (n = 69). Does a RMSE of 2.1 $\mu$g m$^{-3}$ constitute "adequate accuracy"?

7. Line 149: How come the PurpleAir data had to cover 90% of the 24-hour period to be considered "complete," but the FEM data only had to cover 75% of the 24-hour period?

8. Line 166: Why was dewpoint considered in addition to temperature and RH?

9. Lines 168–177: I agree that the model inputs should all be data from the Purple Air. I'm also glad the authors point out that temperature and RH data reported by PurpleAirs are biased high and low, respectively. I don't agree with Reviewer 1 that temperature and RH values reported by PurpleAirs shouldn't be used just because they're inaccurate. Your model does not assume they're accurate; it just assumes that they're correlated with the actual temperature and RH (see Figure S6 in Magi et al., 2020 and Figures S12–S13 in Tryner et al., 2020). I'm not sure I agree that the temperature and RH reported by the BME280 reflect what the aerosol experiences during measurement. Inaccuracies in the PurpleAir temperature and RH data arise because there is no convective or forced airflow over the BME280 to dissipate heat from the breakout board or other electronic components inside the housing. The inlets to the PMS5003 sensors are at the bottom of the housing, and there is active airflow through the PMS5003 sensors.

10. I agree with Reviewer 3 that correlation between input variables warrants further consideration. Can the authors add a correlogram to the supplement to indicate the extent to which different variables were correlated (lines 179–182)? Might the sensitivity of the model coefficients that the authors discuss on lines 347–349 be due to correlation between the independent variables? Are temperature and RH correlated? Could that be why temperature does not reduce error and bias in the nationwide dataset (line 362)—because it essentially provides the same information as RH?

11. Lines 202–206: Were these analyses performed using the model fit to the full dataset (Equation 8)? Or were they performed using the cross-validation methods described on lines 187–195? If the model fit to the full dataset was used, how do the authors justify that choice? Wouldn't the model be expected to almost always predict the AQI correctly for the dataset it was fit to?

12. Lines 220–231: It's great to see this thorough discussion of quality assurance criteria!

13. Line 239: Were these summary statistics calculated before reducing the size of the Iowa dataset and, if so, did any of these values change after?

14. Line 242: By "10,907 days," do the authors mean 10,907 pairs of PA/FM 24-hour PM$_{2.5}$ concentrations? Were multiple data points from a single site on a single day (either multiple PurpleAirs or multiple regulatory monitors) treated as one day or as multiple days? Please clarify. This comment also applies to Table 1. There certainly weren't 3762 days between 11/29/2017 and 12/29/2019.

15. Lines 310–312: I agree that it's more appropriate to use the cf_1 data in your model instead of the cf_atm data displayed on the PurpleAir website. The cf_1 data explained more variance than cf_atm data and cf_atm data are known to be nonlinear at concentrations above 25 $\mu$g m$^{-3}$.

16. Line 335: Does the "linear correction" refer to Equation 3?

17. Lines 335–337: Is there a typo here? The text says that MBE drops from 3.3 to 0 $\mu$g m$^{-3}$ for LOBD and from 4.2 to 0 $\mu$g m$^{-3}$ for LOSO, but in Figure 4 it looks like the opposite: MBE drops from 4.2 to 0 $\mu$g m$^{-3}$ for LOBD and from 3.3 to 0 $\mu$g m$^{-3}$ for LOSO.

18. Sections 3.4 and 3.5: Did the authors look at whether/how their results varied if only FRM measurements were used as the reference or if only measurements from a single type of FEM (BAM, spectrometer, TEOM) were used as the reference? It seems like the authors' large dataset could be used to investigate this question, which might inform others who try to correct PM sensors by collocating them with regulatory monitors. Groups conducting smaller collocation studies that span a small geographic area might only have access to a single type of FEM, so it would be good to know how that might affect their results.

19. Lines 443–449: Do the authors think the lack of an RH term in the AQ&U model is the reason it overestimates PM$_{2.5}$ concentrations for the authors' dataset? RH is typically low in Salt Lake City, while the authors of this study have data from several sites in more humid locations.

20. Lines 489–490: Is the BME280 sensor inside the PurpleAir expected to fail before the PMS5003 sensors? Is there evidence (in the literature or from the authors' experience) that this has occurred in the past? If not, why is this concern emphasized?

21. Lines 512–515: Did the authors provide guidance to the air monitoring agencies on how to appropriately site the PurpleAirs? If so, can the authors relay that guidance in Section 2.1? It might be helpful to PurpleAir users. If not, would the authors like to comment on what they would consider ideal siting and/or provide examples of siting pitfalls? I feel like this might be appropriate given the emphasis on quality assurance elsewhere in the manuscript.

22. Line 530: On lines 150–151, the authors note that they included exceptional event days in their analysis. Did/can the authors identify specific events like dust storms and

wildfires with the help of site operators? Have the authors considered looking at the performance of their model specifically during extreme events? I think the results would be of interest to readers.

23. Table 2: Do these metrics describe the performance of Equation 8? Equation 8 was fit to the data described by the "AB, completeness, problem sensors" criteria, correct? If so, I'm not sure it's appropriate to evaluate a model using the same data used to fit the model and then compare that performance to other datasets that the model was not fit to (rows 1–6).

24. Figure 4: I'm not sure that the ALL column adds useful information to this figure (for the reason given on lines 341–342).

ORGANIZATION

A lot of material is presented in a way that seems disjointed. One way to remedy this issue would be to consistently describe methods in the methods section and results in the results section. Another option is to organize the sections after the introduction as Data Collection, Quality Assurance, Model Development, and Model Evaluation instead of using conventional Methods and Results sections.

1. Lines 102–109: It's great that the authors (i) point out the need to look for erroneous temperature and relative humidity values during the quality assurance process and (ii) describe the erroneous values that they observed! This information will help PurpleAir users look for erroneous temperature and relative humidity data to exclude from their own analyses. However, I think it would be more appropriate for the information on the nature and frequency of the erroneous temperature and RH values to appear along with the information on how many data points were excluded due to disagreement between the channel A and B PM$_{2.5}$ concentrations (Section 3.1).

2. Lines 116–120: The results shown in Figure 1 are discussed here and in Section 3.1. I would prefer to see this discussion all in one place.

3. Lines 121–127: If the "within 5 $\mu$g m$^{-3}$ or 61%" check was performed on the 2-minute/80-second averages, I think it would make more sense for this text to appear before the paragraph describing the completeness threshold. If the "within 5 $\mu$g m$^{-3}$ or 61%" check was performed on 24-hour averages, I think this text should remain here.

4. Lines 128–134: I think this text should appear before the paragraph on lines 102–109. The authors should describe how they downloaded the data before describing how they cleaned and averaged the data.

5. Lines 135–142: This material should appear after the first paragraph in Section 2.2.1 so that the authors' description of how they compared channel A and B data for quality assurance will make sense to readers who are not already familiar with Plantower sensors or PurpleAirs.

6. Lines 219–220: I think this sentence should be moved to Section 2.2.1. Can the authors also provide additional information on this collocation? Where did it take place? How many PurpleAirs were collocated? I assume this was only monitors sent out by EPA and not monitors already being operated by air monitoring agencies. How long did the collocation last? What duration were data averaged over before calculating Pearson and Spearman correlations? What duration were data averaged over before comparing the channel A and B PM$_{2.5}$ concentrations?

7. Lines 242–251: I think it would be more appropriate for this paragraph to appear along with the material discussed in Section 2.2.

8. Line 255: I would prefer to see Table S4 in the main manuscript. In Table S4, why is the RH model just called "RH" but the temperature and dew point models are called "+T" and "+D," respectively? Can the models be labeled RH, T, and D or +RH, +T, and +D? Also, why not group Nonlinear RH with RH, +T, and +D? Why not group +RH+T+D with +RH+T, +RH+D, and +D+T?

9. Lines 262–264: I think this sentence should appear with the description of Plantower

data in Section 2.2.1.

10. If the authors want to stick with the conventional "methods" and "results" organization, the text on lines 255–256, 262–272, 276–277, 288–293, 299–300, 306, and 310–329 is technically methods. The models considered should be described in the methods section and then the amount of variance explained/comparison to prior studies should appear in the results section. If the authors wish to use some alternative organization, I think it would make sense to merge Section 2.3 with Section 3.3.

TECHNICAL CORRECTIONS

1. Line 12: Please insert a comma after "sensors" and another after "(PM)".

2. Lines 45–50: This sentence contains five instances of "and" and zero commas, which made it difficult to follow. Please consider rephrasing or splitting it into multiple sentences.

3. Line 53: Insert "(Plantower PMS5003)" after "particle sensors".

4. Line 54: Specify "(Bosch BME280)" instead of "(BME280)".

5. Line 68: "location specific" should be "location-specific".

6. Line 71 and elsewhere: "U.S. wide" should be "U.S.-wide".

7. Line 98: Please insert a comma after "sensors" and another after "B".

8. Lines 113–114: I suggest rewriting this sentence in past tense to be consistent with the rest of the paragraph (change "ensures" to "ensured" and "are" to "were").

9. Lines 135–136: This sentence was difficult to follow. Maybe rephrase as: "The Plantower sensor reports estimated mass of particles with aerodynamic diameters < 1 $\mu$m (PM$_1$), < 2.5 $\mu$m (PM$_{2.5}$), and < 10 $\mu$m (PM$_{10}$)."

10. Lines 145–146: I think "earliest data" should be "date".

11. Line 148: "local time" should be "local standard time," correct?

12. Lines 284–285: I assume "30 and 80%" refers to RH? Can you write "30% and 80% RH" to ensure this is clear?

13. Line 305: Insert a comma after "Lastly".

14. Line 311: "corrections" should be "correction".

15. Line 314: There are two periods at the end of this sentence.

16. Line 406: The authors alternate between "quality assurance," "cleaning," and "quality control" throughout the manuscript. More consistent terminology would be preferable.

17. Line 420–421: Should this sentence read, "In this work, we also excluded three PurpleAir sensors because there was overall poor agreement between the A and B channels even after excluding individual data points."? If not, then maybe I do not understand what the authors are trying to say.

18. Line 562: Should there be an "and" between "correction methods" and "quality assurance methodology"?

19. Line 559–561: Can the authors please revise this sentence? It's quite long and contains some redundant statements.

20. Figure 1: I like how the authors use this figure to illustrate possible data quality issues, but it's really hard to see the data here. Can the authors make the whole figure larger, use smaller strip text so that the plot panels are larger, and use open circles as markers? Can the authors also either reorder the facets so that the PurpleAirs in CA, for example, appear in the order CA1, CA2, CA3, CA4, CA5, CA6, CA7, CA10, CA11, CA12, CA13, CA15, CA16, CA17, CA18, CA19? This could be accomplished by labeling "CA1" as "CA01" or by reordering the factor associated with these labels.

21. Table S7: There is a broken reference at the end of the caption.

---

## Author Comment (AC1) · 12 Mar 2021

We thank the reviewer for their time reviewing our manuscript and their feedback. We have taken the time to make a number of significant improvements to the manuscript based on the reviewers' comments. We have included responses in blue below.

Anonymous Referee #3

The manuscript presents a development of a generalized correction equation for the PurpleAir PM$_{2.5}$ sensors in the USA. The used data set includes 39 different sites in 16 states and 50 sensor units altogether. The number of recorded datapoints was 12,635 (24h-average, from 8 to 3762 per state). Only sites which had reference level data available no further than 50m away were included in the analysis.

Major comment:

The chosen analysis approach does not utilize the data as well as it could, and there is a strong case to be made that the generalized correction equation is not representative: in fact, a single generalized and true correction equation is most probably impossible to form due to the site-specific differences in aerosol composition. Is it not obvious that two different sites entailing two completely different environments (e.g. rural background vs. urban city center) are not representative of each other? To add, the analysis does not consider possible seasonality, which may have a significant impact on aerosol composition: the shortest sensor data is composed of only 8 data points (MT, Table 1). When these issues are coupled with the notion that not all the factors affecting the sensor behavior are fully understood (line 296), generalizations should be made with extreme caution. In my opinion, the only way to address this issue is to generate correction equations for a few, most generalizable environments (e.g. urban, urban background, detached housing area, regional background) with known aerosol sources. This would also lend for a further investigation regarding the underlying reasons affecting sensor responses.

While we acknowledge that future work could dig more into further reducing error and specific projects could make more accurate corrections for small areas that is not the objective of this project. While site- or location-specific corrections may have merit, this dataset, which to our knowledge is the most comprehensive published in the literature to date, is insufficient to generate these types of localized corrections for use nationally. We believe our dataset is comprehensive enough, including the range of meteorological conditions and aerosol compositions, to build a generalized correction for the entire U.S. that is fairly accurate and able to be implemented easily. We have shown in this paper that this generalized correction is similar to past corrections in the literature suggesting that it is representative of previously studied areas as well. The developed correction equation is valuable for U.S.-wide correction of PurpleAir sensors. If we tried to apply a correction based on environment or seasonality across the U.S. it would be impossible since this information is unknown and there are too many gaps in our dataset.

Technical comment:

In multiple linear regression independent variables should be independent. In my opinion, the fact that previous studies have misused linear regression does not warrant for a new research to continue misusing it. It reinforces the bad habit and undermines the quality and significance of the whole research line of low-cost aerosol sensors. Besides the relative humidity and temperature analysis, this applies also for the analysis regarding the binned particle counts (line 270).

> We have removed all equations that have strongly correlated additive terms in the discussion. See the updated set of equations considered in Table 2:

**Table 2. Correction equation forms considered, and the root mean squared error (RMSE). The best performing model from each increasing complexity (as indicated with *) was validated using withholding in the next sections (Sections 4.3.2 and 4.4).**

| Name | Eqn | RMSE ($\mu g\ m^{-3}$) | RMSE ($\mu g\ m^{-3}$) |
|---|---|---|---|
| | | (cf_1) | (cf_atm) |
| linear | $PA = PM_{2.5}*s_1 + b$ | 2.88* | 3.01 |
| +RH | $PA = s_1*PM_{2.5} + s_2*RH + i$ | 2.52* | 2.59 |
| +T | $PA = s_1*PM_{2.5} + s_2*T + i$ | 2.84 | 2.96 |
| +D | $PA = s_1*PM_{2.5} + s_2*D + i$ | 2.86 | 2.99 |
| +RH*T | $PA = s_1*PM_{2.5} + s_2*RH + s_3*T + s_4*RH*T + i$ | 2.52 | 2.60 |
| +RH*D | $PA = s_1*PM_{2.5} + s_2*RH + s_3*D + s_4*RH*D + i$ | 2.52 | 2.60 |
| +D*T | $PA = s_1*PM_{2.5} + s_2*D + s_3*T + s_4*D*T + i$ | 2.51* | 2.61 |
| +RH*T*D | $PA = s_1*PM_{2.5} + s_2*RH + s_3*T + s_4*D + s_5*RH*T + s_6*RH*D + s_7*T*D + s_8*RH*T*D + i$ | 2.48* | 2.57 |
| PM*RH | $PA = s_1*PM_{2.5} + s_2*RH + s_3*RH*PM_{2.5} + i$ | 2.48* | 2.53 |
| PM*T | $PA = s_1*PM_{2.5} + s_2*T + s_3*T*PM_{2.5} + i$ | 2.84 | 2.96 |
| PM*D | $PA = s_1*PM_{2.5} + s_2*D + s_3*D*PM_{2.5} + i$ | 2.86 | 3.00 |
| PM* Nonlinear RH | $PA = s_1*PM_{2.5} + s_2\frac{RH^2}{(1-RH)}*PM_{2.5} + s_3*\frac{RH^2}{(1-RH)} + i$ | 2.86 | 2.99 |
| PM*RH*T | $PA = s_1*PM_{2.5} + s_2*RH + s_3*T + s_4*PM_{2.5}*RH + s_5*PM_{2.5}*T + s_6*RH*T + s_7*PM_{2.5}*RH*T + i$ | 2.46* | 2.53 |
| PM*RH*D | $PA = s_1*PM_{2.5} + s_2*RH + s_3*D + s_4*PM_{2.5}*RH + s_5*PM_{2.5}*D + s_6*RH*D + s_7*PM_{2.5}*RH*D + i$ | 2.54 | 2.57 |
| PM*T*D | $PA = s_1*PM_{2.5} + s_2*T + s_3*D + s_4*PM_{2.5}*T + s_5*PM_{2.5}*D + s_6*T*D + s_7*PM_{2.5}*T*D + i$ | 2.52 | 2.63 |
| PM*RH*T*D | $PA = s_1*PM_{2.5} + s_2*RH + s_3*T + s_4*D + s_5*PM_{2.5}*RH + s_6*PM_{2.5}*T + s_7*T*RH + s_8*PM_{2.5}*D + s_9*D*RH + s_{10}*D*T + s_{11}*PM_{2.5}*RH*T + s_{12}*PM_{2.5}*RH*D + s_{13}*PM_{2.5}*D*T + s_{14}*D*RH*T + s_{15}*PM_{2.5}*RH*T*D\ i$ | 2.42* | 2.51 |

We have also added correlograms and discussion of them to the text (see response to reviewer 2 above).

Another misuse particularly characteristic for low-cost sensor studies is the use of R2 as goodness-of-fit indicator in nonlinear regression. R2 is not valid for nonlinear regression (line 295).

We have updated the text to consider RMSE instead of $R^2$ (Table 2).

Recommendation: reject

It is our hope that the extensive edits and improvements to the updated manuscript in response to the numerous thoughtful comments from all reviewers will change this recommendation.

---

## Author Comment (AC2) · 12 Mar 2021

We thank the reviewers for their time reviewing our manuscript and their feedback. We have taken the time to make a number of significant improvements to the manuscript based on the reviewers' comments. We have included responses in blue below where the line numbers refer to the manuscript with track changes enabled.

Referee #1

The paper Development and Application of a United States wide correction for $PM_{2.5}$ data collected with the PurpleAir sensor by Barkjohn et al. is an evaluation of the PA units using reference sensors for $PM_{2.5}$ values. Overall, the paper is well written and clear, I have some concerns and comments for things that hopefully you will be able to address and clarify

General comments

1. My biggest concern is about the fact you used the T and RH from the PA itself to perform your analysis. Even the company itself does not recommend using these values for ambient conditions for the same reason you mention in your paper. Also, you cite Holder et al. 2020 but even they state that "the PA temperature and RH measurement are interpreted as an internal rather than ambient measurement". You also enhance this comment in line 375.

We acknowledge that the temperature and RH sensors are not ambient but are internal measurements that run typically hotter and dryer than ambient measurements. Past work has shown that these measurements are strongly correlated with reference measurements making them good candidates for inclusion in the correction. Many PurpleAir sensors are located far from meteorological sites, so if we used reference meteorological data this would make our correction difficult to apply at rural sites where sensors have the most potential to increase spatial resolution on the map. In addition, these measurements may actually be more relevant in understanding hygroscopic growth since particles may be warmed and dried inside the sensor before being measured. Our work is showing that we don't need an ambient measurement of RH to improve the PurpleAir measurements but that the RH reported by the sensor is able to improve the measurements. This question was already addressed in the text:

**Lines 392-403:** It is important to note that the meteorological sensor in the PurpleAir sensor is positioned above the particle sensors nestled under the PVC cap resulting in temperatures that are higher (2.7 to 5.3°C) and RH that is drier (9.7% to 24.3%) than ambient conditions (Holder et al., 2020;Malings et al., 2020). In addition, these internal measurements have been shown to be strongly correlated with reference temperature and RH measurements with high precision (Holder et al., 2020;Tryner et al., 2020a;Magi et al., 2019). The well characterized biases and strong correlations between PurpleAir and ambient meteorological parameters mean that the coefficients using these terms in a correction equation account for the differences between the ambient and PurpleAir measured meteorology. Although not as accurate as the reference measurements, the PurpleAir temperature and RH measurements are good candidates for inclusion in a linear model because they are well correlated with reference measurements and may more closely represent the particle drying that occurs inside the sensor. In addition, using onboard measurements and information that would be available for all PurpleAir sensors, allows

us to gather corrected air quality data from all PurpleAirs, even those in remote areas far from other air monitoring or meteorological sites.

2. Information on the location of co-located units (lat/long) will help the community to understand which purple air sensor you used (pubic), also if you could provide their name that will be important, at least for the public one.

Latitude and Longitude information is available from AQS associated with the AQS ID. We have included them in the SI as well for convenience as requested by the reviewer. The public IDs have also been included for the PurpleAirs (See SI Tables S1, S2)

3. Why 50 meters for distance for collocated units, 1 km is not good for that, other works used a distance larger than 50 m for collocated units. I am wondering how many more collocated units you could have gotten if you had a larger distance between units.

Our goal in this work was to leverage the partnerships air monitoring agencies have with the EPA in order to identify true collocations where a PurpleAir was actually running at the air monitoring site as opposed to close by. As stated in the text, 50m was selected to account for the typical footprint of an air monitoring station. This helps us to constrain that most of the error is due to inaccuracies in the sensor as opposed to localized sources or inaccurate location information.

This was addressed in the text on **lines 95-96:** The 50-meter distance was selected because it is large enough to cover the footprint of most AQS sites and small enough to exclude most PurpleAir sensors in close proximity, but not collocated with, an AQS site.

And we have added additional text **lines 103-110:** Much past work using public data from PurpleAir has used public sensors that appear close to a regulatory station on the map (Ardon-Dryer et al., 2020;Bi et al., 2020). However, there is uncertainty in the reported location of PurpleAir sensors as this is specified by the sensor owner. In some cases, sensors may have the wrong location. Known examples include owners who forgot to update the location when they moved, take the sensor inside for periods to check their indoor air quality, or specifically choose an incorrect location to protect their privacy. In addition, without information on local sources of $PM_{2.5}$, it can be unclear how far away is acceptable for a "collocation" since areas with more localized sources will need to be closer to the reference monitor to experience similar $PM_{2.5}$ conditions. By limiting this work to true collocations operated by air monitoring agencies, we eliminate one source of uncertainty. We can conclude that the PurpleAir errors measured in this work are not due to poor siting or localized sources and can focus on other variables that influence error (e.g. RH).

4. In table 1 you used 50 PA units but in Fig 1 you show more than 50, it is confusing. Maybe have in table 1 the full number of units

We have clarified this in the in the caption of Table 1 and have removed the first reference to table 1.

**Table 1.** Summary of the dataset used to generate the U.S-wide PurpleAir correction equation after 3 sensors with large A, B channel discrepancies were removed. $PM_{2.5}$ concentrations from both the FEM or FRM and the PurpleAir (PA), temperature (T) and relative humidity (RH) are summarized as median (min, max).

**Line 116-117:** In total, 53 PurpleAir sensors at 39 unique sites across 16 states were ideal candidates and were initially included in this analysis

5. Why did you remove data from a station like Iowa, why not using as much data as you have as you could have a full range of T ad RH conditions?

As shown in Figure S1 after subsetting the Iowa data, the full range of temperature and relative humidity conditions are still represented. Using the entire Iowa dataset could lead to a correction that is weighted too heavily towards the aerosol and meteorological conditions experienced in Iowa. This is addressed in the text:

**Lines 122-125:** Initially, there were 10,907 pairs of 24-hr averaged collocated data from Iowa which was 55% of the entire collocated dataset. In order to better balance the dataset among the states, and to avoid building a correction model that is weighted too heavily towards the aerosol and meteorological conditions experienced in Iowa, the number of days from Iowa was reduced to equal the size of the California dataset, the state with the next largest amount of data (29% of the entire collocated dataset).

6. PA company record on using the cf_atm so why also testing the other type?

Although the cf_atm may more closely reflect $PM_{2.5}$ concentrations without correction, it was important for this work to explore which correction was more strongly correlated and more linear with reference $PM_{2.5}$ concentrations. It is not clear that PurpleAir comprehensively evaluated which correction would be best suited for the large range of ambient conditions that exists across the United States. Since PurpleAir provides both it seems like there is no real reason to exclude one from consideration. This is addressed in the text:

**Lines 368-370:** For PM, we considered $PM_{2.5}$ concentrations from both the [cf_1] and [cf_atm] data columns as model terms. Previous work has found different columns to be more strongly correlated under different conditions (Barkjohn et al., 2020a;Tryner et al., 2020a).

7. In Section 3.5 you mention that the use of T for correction should be used on a local basis, but don't you think this is an important factor. In your work, you covered locations with wide T conditions from cold Alaska to warm Florida is it possible that only changes in RH can represent the way to correct the PA data across the US?

We did not write that a local T correction "should" be used just that it has been used in past work and that it may address local factors. It is well known that when relative humidity increases, particles undergo hygroscopic growth (dependent on composition and other particle properties), causing them to scatter more light. Since FEM and FRM methods based on a consistent, relatively low RH range (30-40%), this leads to errors in estimating "dry" $PM_{2.5}$. With temperature there is no similar mechanism that is widely understood to impact the performance of optical measurements. It is unclear what is being accounted for when a temperature term is included in a correction for a single location. It could be accounting for different sources during different parts of the year or something else. Although we have a large dataset, we do not have a large enough dataset to build corrections for each region of the U.S. This is addressed in the text:

**Lines 678-690:** This work indicates that only an RH correction is needed to reduce the error and bias in the nationwide dataset. Some previous single site studies found temperature to significantly improve their $PM_{2.5}$ prediction as well (Magi et al., 2019;Si et al., 2020). Humidity has known impacts on the light scattering of particles; no similar principle exists for explaining the influence of temperature on particle light scattering. Instead, the temperature factor may help account for some local seasonal or diurnal patterns in aerosol properties within smaller geographical areas. These more local variations may be why temperature does not substantially reduce error and bias in the nationwide dataset. More work should be done to better understand this influence. These previous models also did not include a term accounting for the collinearity between temperature and relative humidity that may have been present. **Error! Reference source not found.** shows the residual error in each 24-hour corrected PurpleAir $PM_{2.5}$ measurement compared with the temperature, RH, and FRM or FEM $PM_{2.5}$ concentrations. Error has been reduced compared to the raw dataset (Figures S8, and S9) and is unrelated to temperature, RH, and $PM_{2.5}$ variables. Some bias at very low temperature < -12°C and potentially high concentration (> 60 µg m$^{-3}$) may remain, but more data are needed to further understand this relationship.

Also, from Table S2, it seems most of your RH were < 100, only 3 PA units reach RH of 100 is it possible the lack of high RH harm your analysis.

Lines 341-343: There was limited data above 80% RH as measured by the PurpleAir RH sensor likely due to the warmer and dryer conditions inside the PurpleAir as mentioned previously.

8. Would be nice to see the distribution of RH and T from all dataset (not just for Iowa)

Thank you for this suggestion. We have added it to the SI (Figure S5).

[Figure]

Figure S5. Distribution of full dataset after subsetting Iowa

**Comments of figures and tables**

9. Many of your figures are pixelated

> My apologies. I believe this is from the way I export the pdf. I will make sure to export at higher quality for the next draft.

10. Table 1 - I assume PA data on table 1 is uncorrected. What do you mean by the wide PurpleAir correction equation?

> We have hyphenated U.S.-wide consistently throughout the text to clarify this.

11. Fig 1 - The quality of the fig itself is not great it is very pixelated, also $R^2$ values on each plot would have helped to see the comparison between A and B. you should more h=than 50

> I have added pearson correlation (since correlation is discussed in the text instead of $R^2$) to each plot and exported at higher resolution as requested. See the updated figure below:

[Figure]

Figure 1. Comparison of 24-hour averaged PM$_{2.5}$ data from the PurpleAir A and B channels. Excluded data (2.1%) are shown in red and represent data points where channels differed by more than 5 µg m$^{-3}$ and 61%. AK3, CA7, WA5 were excluded from further analysis. Pearson correlation (r) is shown on each plot.

12. Fig 7 - Does the data represent the entire data set or one location?

This is the full dataset. The caption has been updated as follows:

Figure 7. 24-hr AQI categories as measured by the corrected PurpleAir and the FEM or FRM for the full dataset generated with the models built using LOSO withholding.

Specific comments

13. Line 56-60: it seems you have too many references can you reduce the amount or at least separate them into multiple sentences

I sorted them into a few sentences for clarity.

**Lines 56-67:** Previous work has explored the performance and accuracy of PurpleAir sensors under outdoor ambient conditions in a variety of locations across the United States including in Colorado (Ardon-Dryer et al., 2020;Tryner et al., 2020a), Utah (Ardon-Dryer et al., 2020;Kelly et al., 2017;Sayahi et al., 2019), Pennsylvania (Malings et al., 2020), North Carolina (Magi et al., 2019), and in California where the most work has occurred to date (Ardon-Dryer et al., 2020;Bi et al., 2020;Feenstra et al., 2019;Mehadi et al., 2020;Schulte et al., 2020;Lu et al., 2021). Their performance has been explored in a number of other parts of the world as well including in Korea (Kim et al., 2019), Greece (Stavroulas et al., 2020), and Australia (Robinson, 2020). Additional work has been done to evaluate their performance under wildland fire smoke impacted conditions (Bi et al., 2020;Delp and Singer, 2020;Holder et al., 2020), indoors (Wang et al., 2020b), and during laboratory evaluations (Kelly et al., 2017;Kim et al., 2019;Li et al., 2020;Mehadi et al., 2020;Tryner et al., 2020a;Zou et al., 2020a;Zou et al., 2020b). The performance of their dual Plantower PMS5003 laser scattering particle sensors has also been explored in a variety of other commercial and custom built sensor packages (He et al., 2020;Tryner et al., 2019;Kuula et al., 2019;Ford et al., 2019;Si et al., 2020;Zou et al., 2020b;Tryner et al., 2020b).

Line 85: you state that the data was until 2018 but some of your data was for 2019

Since this analysis occurred in 2018 we identified sensors that were online in 2017 and 2018. This section has been rewritten to improve clarity

**Lines 85-95:** Data for this project came from 3 sources: 1) PurpleAir sensors sent out by EPA for collocation to capture a wide range of regions and meteorological conditions, 2) privately operated sensor data volunteered by state, local and tribal (SLT) air monitoring agencies independently operating collocated PurpleAir sensors, and 3) publicly available sensors located near monitoring stations and confirmed as true collocation by air monitoring agency staff. In order to identify publicly available collocated sensors, in August of 2018, a survey of sites with potentially collocated PurpleAir sensors and regulatory $PM_{2.5}$ monitors was performed by identifying publicly available PurpleAir sensor locations within 50 meters of an active EPA Air Quality System (AQS) site reporting $PM_{2.5}$ data in 2017 or 2018.

14. Lines 110-114: the entire section is unclear, I do not understand what you mean, can you rewrite this part to make it clearer

We have clarified this section.

**Lines 260-265:** The 2-minute (or 80-second) $PM_{2.5}$ data were averaged to 24-hours (representing midnight to midnight local standard time). A 90% data completeness threshold was used based on channel A, since both channels were almost always available together (i.e. 80-second averages required at least 0.9*1080 points before 5/30/2019 or 2-minute averages required at least 0.9*720 points after 5/30/2019). This methodology ensured that the averages

used were truly representative of daily averages reported by regulatory monitors. A higher threshold of completeness was used for the PurpleAir data as it likely has more error than FEM or FRM measurements.

15. Line 235: what do you mean - one ran multiple sensors in series

We have clarified the meaning of series in this context:

**Lines 333-335:** Some sites had several PurpleAir sensors running simultaneously (N=9) and one ran multiple sensors in series (i.e. one sensor failed, was removed, and another sensor was put up in its place).

---

## Author Comment (AC3) · 12 Mar 2021

We thank the reviewer for their time reviewing our manuscript and their feedback. We have taken the time to make a number of significant improvements to the manuscript based on the reviewers' comments. We have included responses in blue below where the line numbers refer to the manuscript with track changes enabled.

**Referee #2**

GENERAL COMMENTS The authors present carefully-considered, practical guidance for performing quality assurance checks on PurpleAir data. The authors' model for correcting PurpleAir PM2.5 data benefits from a large training dataset that spans many months and locations (39 sites in 16 U.S. states). Together, the quality assurance guidelines and correction model presented here can help users improve the accuracy of $PM_{2.5}$ data reported by PurpleAirs. There will likely be applications in which this U.S.-wide correction still does not produce sufficiently-accurate estimates of local ambient $PM_{2.5}$ concentrations, but I am not here to argue for letting the perfect be the enemy of the good.

We thank the reviewer for their thoughtful review.

SPECIFIC COMMENTS

1. Lines 79–82: I found these two sentences confusing. Are the PurpleAirs that are part of the "larger project" counted as PurpleAirs sent out by EPA (group 1 on line 77) or PurpleAirs independently operated by air monitoring agencies (group 2 on line 77)? Can the authors clarify how these PurpleAirs fit in? Alternatively, can the authors remove these sentences and simply describe the study as involving PurpleAirs "sent out by EPA to capture a wide range of regions and meteorological conditions" and "other collocated sensors"?

We have clarified these lines as follows:

**Lines 85-88:** Data for this project came from 3 sources: 1) PurpleAir sensors sent out by EPA for collocation to capture a wide range of regions and meteorological conditions, 2) privately operated sensor data volunteered by state, local and tribal (SLT) air monitoring agencies independently operating collocated PurpleAir sensors, and 3) publicly available sensors located near monitoring stations and confirmed as true collocation by air monitoring agency staff.

2. Lines 91–92: The authors state here that their analysis included data from 50 PurpleAirs at 39 sites in 16 states. On lines 87–88, the authors state that their search identified 42 sites in 14 states. Does that mean the 50 PurpleAirs/39 sites/16 states includes both PurpleAirs sent out by EPA and PurpleAirs identified in the search?

Yes. This has been clarified in the text:

**Lines 85-101:** Data for this project came from 3 sources: 1) PurpleAir sensors sent out by EPA for collocation to capture a wide range of regions and meteorological conditions, 2) privately operated sensor data volunteered by state, local and tribal (SLT) air monitoring agencies independently operating collocated PurpleAir sensors, and 3)

publicly available sensors located near monitoring stations and confirmed as true collocation by air monitoring agency staff. In order to identify publicly available collocated sensors, in August of 2018, a survey of sites with potentially collocated PurpleAir sensors and regulatory $PM_{2.5}$ monitors was performed by identifying publicly available PurpleAir sensor locations within 50 meters of an active EPA Air Quality System (AQS) site reporting $PM_{2.5}$ data in 2017 or 2018. The 50-meter distance was selected because it is large enough to cover the footprint of most AQS sites and small enough to exclude most PurpleAir sensors in close proximity, but not collocated with, an AQS site. From a download of all active AQS $PM_{2.5}$ sites and PurpleAir sensor locations on August 20, 2018, 42 unique sites were identified in 14 states. From this list of public PurpleAir sensors potentially collocated with regulatory $PM_{2.5}$ monitors, we reached out to the appropriate SLT air monitoring agency to understand if these units were operated by the air monitoring agency and their interest in partnering in this research effort. If we could not identify the sensor operator of these 42 sensors, or if the sensor was not collocated at the air monitoring station, the sensor was not used in this analysis.

**Lines 116-120:** In total, 53 PurpleAir sensors at 39 unique sites across 16 states were ideal candidates and were initially included in this analysis with data included from September 2017 until January 2020. The supplement contains additional information about each AQS site (Table S1) and each individual sensor (Table S2).

3. Lines 121–127: Was the "within 5 µg m$^{-3}$ or 61%" check performed on the 2- minute/80-second averages or on the 24-hour averages? Was this check performed on the cf_1 data or on the cf_atm data? Please clarify.

It was performed on the 24-hr averaged cf_1 data. This has been addressed in the text as follows:

**Lines 283-392:** Data cleaning procedures were developed using the typical 24-hr averaged agreement between the A and B channels expressed as percent error (Eq. 1).

$$24\text{-hr percent difference} = \frac{(A-B)*2}{(A+B)}$$

(1)

Where A and B are the 24-hr average $PM_{2.5}$ cf_1 concentrations from the A and B channels. 24-hour averaged data points with percent differences larger than two standard deviations (2sd=61%) were flagged for removal.

Section 2: Can the authors include some information on the breakdown of regulatory monitors and reference $PM_{2.5}$ measurements by type (FRM vs. FEM and type of FEM)? Maybe this information could be included in Tables S1–S2 and/or summarized in the text? FEM measurements do not always agree with FRM measurements and some FEMs might be more likely to deviate from FRMs than others. I downloaded the daily $PM_{2.5}$ data from the sites listed in Table S1, over the date ranges listed in Table S2, from the AQS. It looks like there were 17 FRMs, 16 BAMs, 16 spectrometers (T640s), and 6 TEOMs across the 39 sites. It looks like the

breakdown of 24-hour measurements was approximately 10% FRM, 50% BAM, 30% spectrometer, and 10% TEOM.

> Yes, we have included these details:
>
> **Lines 228-231:** The dataset was comprised of data from 21 BAM 1020s or 1022s, 19 Teledyne T640 or T640xs, and 5 TEOM 1405s or 1400s. Sixteen sites had FRM measurements. After excluding part of the Iowa dataset BAM1020's provided the most 24-hr averaged points followed by the T640 and T640x, and the RP2025 (Figure S2). 1/5 of the data came from FRM measurements while the rest came from FEMs (Figure S3).

3. Lines 153–154: It looks like there were 13 sites with both FRM and FEM data. It looks like there were three sites (53-033-0057, 53-061-1007, and 55-087-0009) with two FEM monitors and one site (19-113-0040) with two FEMs plus an FRM. What did the authors do when multiple reference measurements (FRM or FEM) were available for a single 24-hour period at a single site? Were all 24-hour average measurements included as separate data points? Or was just one selected and, if so, which one?

> **Line 231-232:** If daily measurements were collected using two methods both points were included in the analysis.

Lines 154–155 and 470–474: Did the authors look at the agreement between 24- hour average FRM and FEM measurements for the dates and sites where both were available? It might be useful for comparison to the agreement between PA and FM data. For the sites and dates listed in Tables S1–S2, it looks like the RMSE for 24-hour FEM measurements, compared to FRM measurements, was 1.7 µg m−3 for BAMs (n = 4134), 2.1 µg m$^{-3}$ for spectrometers (n = 4405), and 1.1 µg m$^{-3}$ for TEOMs (n = 69). Does a RMSE of 2.1 µg m$^{-3}$ constitute "adequate accuracy"?

> We have included analysis based on EPA's tools to understand FEM and FRM performance. We focused on bias instead of RMSE. The T640s are typically biased high (which is likely what is driving the higher RMSE). We have added additional discussion and justification for using these devices.
>
> **Lines 235-255:** 3.1 FRM and FEM Quality Assurance
>
> The accuracy of the FRM and FEM measurements was considered. In total, Federal Reference Method (FRM) data was used from 13 organizations. The accuracy of these measurements was evaluated using the FRM performance assessments (U.S. EPA, 2020b). They were evaluated using the FRM/FRM precision and bias, the average field blank weight, and the monthly precision. The performance of the FEM monitors was evaluated using the $PM_{2.5}$ continuous monitor comparability assessments (U.S. EPA, 2020a). FEM measurements are compared to simultaneous Federal Reference Method (FRM) measurements. Linear regression is used to calculate a slope, intercept (int), and correlation (R) and the FEM/FRM ratio  is also computed. Based on data quality

objectives, the slope should be between 0.9 and 1.1, intercept between -2 and 2, correlation should be between 0.9 and 1, and the ratio should be within 0.9 and 1.1. The most recent 3 years of available data was used to evaluate each monitor.

Performance data was only available for 10 of the 13 collection agencies (77%, Table S3). All available agencies met the FRM/FRM precision goals. All but one state show negative FRM bias suggesting organization reported FRM $PM_{2.5}$ is biased low by 1-22%. Four of the agencies (40%) only marginally fail the ≤10% bias criteria with bias from -10.1% to -11% . The one organization with more significant bias (-22%) is driven by the difference in a single FRM measurement pair. All sites typically have acceptable field blank weights and monthly average precision within 30%. The performance of all FRM measurements are acceptable for use in developing the PurpleAir U.S.-wide correction.

Of the 46 unique FEM monitors, comparability assessments were only available for 24 monitors (51%, Tables S4,S5). All slopes were within the acceptable range. One intercept was slightly outside the acceptable range (2.35) and 3 correlations were slightly below the acceptable limit (0.86-0.89), however these values have been considered acceptable for this use. Of greater concern is that 10 FEMs had ratios greater than 1.1 up to 1.3 (41% of monitors) and these were all Teledyne T640 or T640x devices (Figure S4). The data from the T640 and T640x make up about 20% of the total dataset and excluding them would reduce the diversity of the dataset. Since these monitors are frequently used for regulatory applications, the performance of all FEM measurements has been considered acceptable for use in developing the PurpleAir U.S.-wide correction.

U.S. EPA PM2.5 Continuous Monitor Comparability Assessments: https://www.epa.gov/outdoor-air-quality-data/pm25-continuous-monitor-comparability-assessments, 2020a.
U.S. EPA PM2.5 Data Quality Dashboard: https://sti-r-shiny.shinyapps.io/QVA_Dashboard/, 2020b.

4. Line 149: How come the PurpleAir data had to cover 90% of the 24-hour period to be considered "complete," but the FEM data only had to cover 75% of the 24-hour period?

This was a somewhat arbitrary distinction however, we felt that a stricter criterion was warranted for the PurpleAir data since we assume it has a higher error. We have added additional discussion of this in the text:

**Lines 264-265:** A higher threshold of completeness was used for the PurpleAir data since there will likely be more error in these measurements than the FEM or FRM measurements.

5. Line 166: Why was dewpoint considered in addition to temperature and RH?

Some past work has found the dewpoint was able to explain error that temperature and RH were not able to explain. This has been added to the text.

Lines 392-393: Dew point was considered since past work has shown that dewpoint can, in some cases, explain error unexplained by temperature or RH (Mukherjee et al., 2019;Malings et al., 2020).

6. Lines 168–177: I agree that the model inputs should all be data from the Purple Air. I'm also glad the authors point out that temperature and RH data reported by PurpleAirs are biased high and low, respectively. I don't agree with Reviewer 1 that temperature and RH values reported by PurpleAirs shouldn't be used just because they're inaccurate. Your model does not assume they're accurate; it just assumes that they're correlated with the actual temperature and RH (see Figure S6 in Magi et al., 2020 and Figures S12–S13 in Tryner et al., 2020). I'm not sure I agree that the temperature and RH reported by the BME280 reflect what the aerosol experiences during measurement. Inaccuracies in the PurpleAir temperature and RH data arise because there is no convective or forced airflow over the BME280 to dissipate heat from the breakout board or other electronic components inside the housing. The inlets to the PMS5003 sensors are at the bottom of the housing, and there is active airflow through the PMS5003 sensors.

We have removed the objected sentence and added the citations to Tryner and Magi in this location.

Lines 391-395: It is important to note that the meteorological sensor in the PurpleAir sensor is positioned above the particle sensors nestled under the PVC cap resulting in temperatures that are higher (2.7 to 5.3°C) and RH that is drier (9.7% to 24.3%) than ambient conditions (Holder et al., 2020;Malings et al., 2020) but which may be closer to what is experienced by the aerosol during measurement. In addition, these internal measurements have been shown to be strongly correlated with reference temperature and RH measurements with high precision (Holder et al., 2020;Tryner et al., 2020a;Magi et al., 2019).

10. I agree with Reviewer 3 that correlation between input variables warrants further consideration. Can the authors add a correlogram to the supplement to indicate the extent to which different variables were correlated (lines 179–182)? Might the sensitivity of the model coefficients that the authors discuss on lines 347–349 be due to correlation between the independent variables? Are temperature and RH correlated? Could that be why temperature does not reduce error and bias in the nationwide dataset (line 362)—because it essentially provides the same information as RH?

We appreciate the reviewer's suggestion and have added a correlogram to the SI and additional discussion to the text. We have also removed the consideration of models with moderately to strongly correlated additive terms throughout (see the response to reviewer 3's comments)

Lines 357-367: In a multiple linear regression, all independent variables should be independent; however, much previous work has used models that incorporate additive temperature, RH, and dewpoint terms that are not independent (Magi et al., 2019;Malings

et al., 2020). We have not considered these models and have considered models with interaction terms (i.e. RH*T*PM$_{2.5}$) to account for inter-dependence between terms instead. Strong correlations (r ≥ ±0.7) are shown between the 24-hr averaged FEM or FRM PM$_{2.5}$, PurpleAir estimated PM$_{2.5}$ (cf_1and cf_atm), and each binned count (Figure S6). Since the binned counts include all particles greater than a certain size, we also consider the correlation between the delta of each bin (e.g. particles >0.3 µm – particles >0.5 µm=particles 0.3-0.5 µm). The delta bin counts were still moderately to strongly correlated (r=0.6-1) with the weakest correlation seen between the smallest and largest bins (Figure S7). Moderate correlations (r = ±0.4-0.6) are seen between temperature, RH, and dewpoint. Weak correlations (r ≤ ±0.2) are seen between the PM variables (i.e. PM$_{2.5}$ and bin variables) and environmental variables (i.e. temperature, RH, and Dewpoint). The correlation between variables was considered when considering model forms.

[Figure]

Figure S6. Correlogram showing the pearson correlation (Corr) between considered correction input variables.

[Figure]

Figure S7. Correlogram showing the pearson correlation (Corr) between considered correction input variables including the difference between bins (e.g. b0.3to0.5 = bin<0.3 - bin<0.5).

11. Lines 202–206: Were these analyses performed using the model fit to the full dataset (Equation 8)? Or were they performed using the cross-validation methods described on lines 187–195? If the model fit to the full dataset was used, how do the authors justify that choice? Wouldn't the model be expected to almost always predict the AQI correctly for the dataset it was fit to?

> The reviewer brings up a good point. We have instead plotted the data generated from LOSO withholding.
>
> **Lines 736-739:** Lastly, we summarize the performance of the sensors across the U.S. using the U.S. daily AQI categories (Federal Register, 1999). For this analysis we use the data corrected using the LOSO withholding where a final correction is built for all but one state and then applied to the withheld state.

This allows us to better understand how the correction will perform in locations not included in our analysis.

[Figure]

Figure 7. 24-hr AQI categories as measured by the corrected PurpleAir and the FEM or FRM for the full dataset generated with the models built using LOSO withholding.

12. Lines 220–231: It's great to see this thorough discussion of quality assurance criteria!

Thank you!

13. Line 239: Were these summary statistics calculated before reducing the size of the Iowa dataset and, if so, did any of these values change after?

**Line 350-352:** These summary statistics were calculated after selecting a random subset of the Iowa data. The median of the Iowa dataset increased from 7 to 10 µg m$^{-3}$ after subsetting since more of the high concentration data was conserved.

14. Line 242: By "10,907 days," do the authors mean 10,907 pairs of PA/FM 24-hour PM$_{2.5}$ concentrations? Were multiple data points from a single site on a single day (either multiple PurpleAirs or multiple regulatory monitors) treated as one day or as multiple days? Please clarify. This comment also applies to Table 1. There certainly weren't 3762 days between 11/29/2017 and 12/29/2019.

**Line 122:** Initially, there were 10,907 pairs of 24-hr averaged collocated data from Iowa

**Line 231-232:** If daily measurements were collected using two methods both points were included in the analysis.

**Table 1:** # of  points

15. Lines 310–312: I agree that it's more appropriate to use the cf_1 data in your model instead of the cf_atm data displayed on the PurpleAir website. The cf_1 data explained more variance than cf_atm data and cf_atm data are known to be nonlinear at concentrations above 25 $\mu g\ m^{-3}$.

Thank you.

16. Line 335: Does the "linear correction" refer to Equation 3?

Yes, this has been updated in the text

**Line 665-666:** when applying a linear correction (Eq. 4).

17. Lines 335–337: Is there a typo here? The text says that MBE drops from 3.3 to 0 $\mu g\ m^{-3}$ for LOBD and from 4.2 to 0 $\mu g\ m^{-3}$ for LOSO, but in Figure 4 it looks like the opposite: MBE drops from 4.2 to 0 $\mu g\ m^{-3}$ for LOBD and from 3.3 to 0 $\mu g\ m^{-3}$ for LOSO.

Yes, thank you for catching this typo we have updated in the text.

**Lines 666-667:** Using LOBD, the MBE across withholding runs drops significantly from 4.2 to 0 $\mu g\ m^{-3}$ with a similar significant drop, from 2.8 to 0 $\mu g\ m^{-3}$

18. Sections 3.4 and 3.5: Did the authors look at whether/how their results varied if only FRM measurements were used as the reference or if only measurements from a single type of FEM (BAM, spectrometer, TEOM) were used as the reference? It seems like the authors' large dataset could be used to investigate this question, which might inform others who try to correct PM sensors by collocating them with regulatory monitors. Groups conducting smaller collocation studies that span a small geographic area might only have access to a single type of FEM, so it would be good to know how that might affect their results.

We have added a brief discussion of this in the text.

**Lines 695-710:** 5.3.1 The influence of FEM and FRM type

We briefly considered whether the use of both FEM and FRM measurements influenced these results. When sub-setting the data to develop models using the 24-hr averaged $PM_{2.5}$ data from only the FEM versus only the FRM, only the coefficient for the PA slope term changed. The coefficient was slightly larger for FEM measurement (0.537) and smaller for FRM measurements (0.492). Although the coefficients are significantly different (p<0.05) they are within 10% leading to little difference in the interpretation of PurpleAir $PM_{2.5}$ measurements. We briefly considered whether the FEM coefficient was driven by the T640s and found that if we build this model excluding all T640 and T640x data, it is not significantly different (0.53). Concerns about error between different types of FEM measurements cannot be explored using this dataset. Further, FEM instruments are not randomly distributed across the U.S. but rather clustered at sites operated by the same air agency. Future work and a more concerted effort may be needed to explore this issue. Overall, the accuracy of all these FEM and FRM methods have been determined accurate enough for regulatory purposes and so we have used all to determine our U.S.-

wide correction. Although FRM measurements are the gold standard, using only FRM measurements would have severely limited our dataset. In addition, the use of FEM measurements will be important in future work to explore the performance of this model correction at higher time resolutions. At higher time resolutions, the noise and precision between different FEMs may impact perceived performance and future work should further explore this.

19. Lines 443–449: Do the authors think the lack of an RH term in the AQ&U model is the reason it overestimates PM$_{2.5}$ concentrations for the authors' dataset? RH is typically low in Salt Lake City, while the authors of this study have data from several sites in more humid locations.

We have added an additional sentence to address this

**Line 809-811:** Since RH is typically low in Salt Lake City this may lead to some of the overestimate in using this equation in more humid parts of the country.

20. Lines 489–490: Is the BME280 sensor inside the PurpleAir expected to fail before the PMS5003 sensors? Is there evidence (in the literature or from the authors' experience) that this has occurred in the past? If not, why is this concern emphasized?

It may fail independently of the PMS5003. We saw this occasionally in our dataset and have added additional discussion

**Lines 267-280:** 3.2.2 PurpleAir Temperature and RH errors
For correction model development, it was important to start with the most robust dataset possible. In the 2-minute or 80-second data, occasionally, an extremely high temperature (i.e. 2147483447) or an extremely low temperature (i.e. -224 or -223) was reported, likely due to electrical noise or a communication error between the temperature sensor and the PurpleAir microcontroller. The high error occurred in 24 of 53 sensors but occurred infrequently (34 instances in ~$10^7$ points total) while the low error impacted only 2 sensors (1% of the full dataset). Temperature values above 540°C (1000°F) were excluded before calculating daily averages since temperature errors were extreme and easily detected above this level. Similarly, the RH sensor occasionally read 255%; this problem was experienced by each sensor at least once but still occurred infrequently (1083 points out of ~$10^7$ total). No other values were found outside 0-100% in the 2-minute or 80-second data before averaging. These points were removed from the analysis before 24-hr averaging.

Missing temperature or RH impacted only 2% of the dataset (184 points) with 8 sensors having one to four 24-hr averages with missing temperature or RH. One sensor, WI4, had 167 days (90%) without temperature data. Most of the available temperature data was recorded in the first few weeks of operation. It is unclear what caused the temperature data to be missing. All 184 points were missing temperature but only 17 were also missing RH (0.2% of full dataset).

21. Lines 512–515: Did the authors provide guidance to the air monitoring agencies on how to appropriately site the PurpleAirs? If so, can the authors relay that guidance in Section 2.1? It might be helpful to PurpleAir users. If not, would the authors like to comment on what they would consider ideal siting and/or provide examples of siting pitfalls? I feel like this might be appropriate given the emphasis on quality assurance elsewhere in the manuscript.

> Siting guidance was not explicitly outlined for the agencies, but additional discussion has been added to the text.

> **Lines 111-115:** When EPA provided PurpleAir sensors to air monitoring agencies, EPA suggested that they be deployed with similar siting criteria as regulatory monitors. Some sites had space and power limitations to consider but trained technicians cited sensors allowing adequate unobstructed airflow. In many cases, sensors were attached to the top rung of the railings at the monitoring shelters where they were within a meter or so of other inlet heights and within 3 meters or so of the other instrument inlets.

22. Line 530: On lines 150–151, the authors note that they included exceptional event days in their analysis. Did/can the authors identify specific events like dust storms wildfires with the help of site operators? Have the authors considered looking at the performance of their model specifically during extreme events? I think the results would be of interest to readers.

> We agree that this would be of interest. We are in the process of collecting a more detailed dataset of exceptional events and plan to publish more detail on smoke and dust impacted times in future publications.

23. Table 2: Do these metrics describe the performance of Equation 8? Equation 8 was fit to the data described by the "AB, completeness, problem sensors" criteria, correct? If so, I'm not sure it's appropriate to evaluate a model using the same data used to fit the model and then compare that performance to other datasets that the model was not fit to (rows 1–6).

> We acknowledge the concerns of the reviewer. This is a question that could be explored in much greater detail in future work. We don't want to complicate the message of the paper, so we have made this discussion more general and moved the discussion to the SI (section 3, Table S7) with an acknowledgement of the limitations of our methods. We have removed the table that was in the text and have only left the detailed table that was in the SI (Table S7).

> **Lines 323-330:** 3.2.4 Importance of PurpleAir Quality Control procedures

> This work did not seek to optimize data cleaning procedures to balance data retention with data quality, instead it focused on generating a best-case dataset from which to build a model. However, the removal of outlier points based on the difference between the A and B channels appears to reduce the errors most strongly (Supplement section 3, Table S7) when compared to removing incomplete daily averages or removing problematic sensors. Since both channels are needed for comparison, it makes sense to average the A and B channels to improve the certainty on the measurement. The data completeness

control provides less benefit and may not be needed for all future applications of these correction methods. In addition, sensors with systematic offsets were uncommon and did not largely impact the overall accuracy, so the A and B channel comparison on the 24-hour averaged data points (e.g. 5 µg m$^{-3}$ and 61%) may be sufficient.

24. Figure 4: I'm not sure that the ALL column adds useful information to this figure (for the reason given on lines 341–342).

We have removed this column from the figure.

[Figure]

**Figure 1.** Performance statistics including mean bias error (MBE) and mean absolute error (MAE) are shown by correction method (0-5), where each point in the boxplot is the performance for either a 12-week period excluded from correction building ("LOBD"), or a single state excluded from correction building ("LOSO").

ORGANIZATION

A lot of material is presented in a way that seems disjointed. One way to remedy this issue would be to consistently describe methods in the methods section and results in the results section. Another option is to organize the sections after the introduction as Data Collection, Quality Assurance, Model Development, and Model Evaluation instead of using conventional Methods and Results sections.

> We thank the reviewer for their detailed feedback, and we have reorganized the paper into 2) Data Collection, 3) Quality Assurance, 4) Model Development, 5) Model Evaluation

1.Lines 102–109: It's great that the authors (i) point out the need to look for erroneous temperature and relative humidity values during the quality assurance process and (ii) describe the erroneous values that they observed! This information will help PurpleAir users look for erroneous temperature and relative humidity data to exclude from their own analyses. However, I think it would be more appropriate for the information on the nature and frequency of the erroneous temperature and RH values to appear along with the information on how many data points were excluded due to disagreement between the channel A and B $PM_{2.5}$ concentrations (Section 3.1).

> We have moved this discussion to section 3.2.2 PurpleAir Temperature and RH errors (under section 3.2 PurpleAir Quality Assurance & Data Cleaning)

2.      Lines 116–120: The results shown in Figure 1 are discussed here and in Section 3.1. I would prefer to see this discussion all in one place.
> We have combined this discussion in section 3.2.3

3. Lines 121–127: If the "within 5 µg m$^{-3}$ or 61%" check was performed on the 2- minute/80-second averages, I think it would make more sense for this text to appear before the paragraph describing the completeness threshold. If the "within 5 µg m$^{-3}$ or 61%" check was performed on 24-hour averages, I think this text should remain here.

> This occurred on the 24-hr averaged data. It has been clarified in the text and appears in section 3.2.3

3.      Lines 128–134: I think this text should appear before the paragraph on lines 102– 109. The authors should describe how they downloaded the data before describing how they cleaned and averaged the data.
> We have moved this paragraph as suggested.

> **Now section 2.2.1 lines 150-156:** When a PurpleAir sensor is connected to the internet, data is sent to PurpleAir's data repository on ThingSpeak. Users can choose to make their data publicly viewable (public) or control data sharing (private). Agencies with privately

reporting sensors provided application programming interface (API) keys so that data could be downloaded. PurpleAir PA-II-SD models can also record data offline on a microSD card; however, these offline data appeared to have time stamp errors from internal clocks that drift without access to the frequent time syncs available with access to WiFi so they were excluded from this project. Data were downloaded from the ThingSpeak API using Microsoft PowerShell at the native 2-minute or 80-second time resolution and were saved as csv files that were processed and analysed in R (R Development Core Team, 2019).

4. Lines 135–142: This material should appear after the first paragraph in Section 2.2.1 so that the authors' description of how they compared channel A and B data for quality assurance will make sense to readers who are not already familiar with Plantower sensors or PurpleAirs.

We have moved this information to section 2.2.1

5. Lines 219–220: I think this sentence should be moved to Section 2.2.1. Can the authors also provide additional information on this collocation? Where did it take place? How many PurpleAirs were collocated? I assume this was only monitors sent out by EPA and not monitors already being operated by air monitoring agencies. How long did the collocation last?

This was not a thorough evaluation so this sentence has been removed from the paper.

What duration were data averaged over before calculating Pearson and Spearman correlations? What duration were data averaged over before comparing the channel A and B PM2.5 concentrations?

This was all done at 24-hr averages

**Lines 283-291:** The two Plantower sensors within the PurpleAir sensor (channels A and B) can be used to check the consistency of the data reported. All comparisons in this work have occurred at 24-hour averages. Anecdotal evidence from PurpleAir suggests some disagreements may be caused by spiders, insects, or other minor blockages that may resolve on their own. Data cleaning procedures were developed using the typical 24-hr averaged agreement between the A and B channels expressed as percent error (Eq. 1).

$$24\text{-hr percent difference}=\frac{(A-B)*2}{(A+B)}$$

(1)

Where A and B are the 24-hr average PM$_{2.5}$ cf_1 concentrations from the A and B channels. 24-hour averaged data points with percent differences larger than two standard deviations (2sd=61%) were flagged for removal.

**Lines 315-317:** Six of these 10 sensors had ≤4% of the data removed by data cleaning steps and their Pearson correlation, on 24-hour averages, improved to ≥0.98 (from r < 0.7) suggesting that the low correlation was driven by a few outlier points.

7. Lines 242–251: I think it would be more appropriate for this paragraph to appear along with the material discussed in Section 2.2.

This was moved to section 2.1.1 (lines 122-132)

7. Line 255: I would prefer to see Table S4 in the main manuscript. In Table S4, why is the RH model just called "RH" but the temperature and dew point models are called "+T" and "+D," respectively? Can the models be labeled RH, T, and D or +RH, +T, and +D? Also, why not group Nonlinear RH with RH, +T, and +D? Why not group +RH+T+D with +RH+T, +RH+D, and +D+T?

> We have moved this to the main text and have considered different models to address reviewer 3's comments. We group nonlinear RH with the *PM models since it is multiplied by PM.

**Table 2. Correction equation forms considered and the root mean squared error (RMSE). The best performing model from each increasing complexity (as indicated with *) was validated using withholding in the next sections (Section 5).**

| Name | Eqn | RMSE ($\mu g\ m^{-3}$) | RMSE ($\mu g\ m^{-3}$) |
|---|---|---|---|
| | | (cf_1) | (cf_atm) |
| linear | $PA = PM_{2.5}*s_1 + b$ | 2.88* | 3.01 |
| +RH | $PA = s_1*PM_{2.5} + s_2*RH + i$ | 2.52* | 2.59 |
| +T | $PA = s_1*PM_{2.5} + s_2*T + i$ | 2.84 | 2.96 |
| +D | $PA = s_1*PM_{2.5} + s_2*D + i$ | 2.86 | 2.99 |
| PM* Nonlinear RH | $PA = s_1*PM_{2.5} + s_2\frac{RH^2}{(1-RH)}*PM_{2.5} + s_3*\frac{RH^2}{(1-RH)} + i$ | 2.86 | 2.99 |
| +RH*T | $PA = s_1*PM_{2.5} + s_2*RH + s_3*T + s_4*RH*T + i$ | 2.52 | 2.60 |
| +RH*D | $PA = s_1*PM_{2.5} + s_2*RH + s_3*D + s_4*RH*D + i$ | 2.52 | 2.60 |
| +D*T | $PA = s_1*PM_{2.5} + s_2*D + s_3*T + s_4*D*T + i$ | 2.51* | 2.61 |
| +RH*T*D | $PA = s_1*PM_{2.5} + s_2*RH + s_3*T + s_4*D + s_5*RH*T + s_6*RH*D + s_7*T*D + s_8*RH*T*D + i$ | 2.48* | 2.57 |
| PM*RH | $PA = s_1*PM_{2.5} + s_2*RH + s_3*RH*PM_{2.5} + i$ | 2.48* | 2.53 |
| PM*T | $PA = s_1*PM_{2.5} + s_2*T + s_3*T*PM_{2.5} + i$ | 2.84 | 2.96 |
| PM*D | $PA = s_1*PM_{2.5} + s_2*D + s_3*D*PM_{2.5} + i$ | 2.86 | 3.00 |
| PM*RH*T | $PA = s_1*PM_{2.5} + s_2*RH + s_3*T + s_4*PM_{2.5}*RH + s_5*PM_{2.5}*T + s_6*RH*T + s_7*PM_{2.5}*RH*T + i$ | 2.46* | 2.53 |
| PM*RH*D | $PA = s_1*PM_{2.5} + s_2*RH + s_3*D + s_4*PM_{2.5}*RH + s_5*PM_{2.5}*D + s_6*RH*D + s_7*PM_{2.5}*RH*D + i$ | 2.54 | 2.57 |
| PM*T*D | $PA = s_1*PM_{2.5} + s_2*T + s_3*D + s_4*PM_{2.5}*T + s_5*PM_{2.5}*D + s_6*T*D + s_7*PM_{2.5}*T*D + i$ | 2.52 | 2.63 |
| PM*RH*T*D | $PA = s_1*PM_{2.5} + s_2*RH + s_3*T + s_4*D + s_5*PM_{2.5}*RH + s_6*PM_{2.5}*T + s_7*T*RH + s_8*PM_{2.5}*D + s_9*D*RH + s_{10}*D*T + s_{11}*PM_{2.5}*RH*T + s_{12}*PM_{2.5}*RH*D + s_{13}*PM_{2.5}*D*T + s_{14}*D*RH*T + s_{15}*PM_{2.5}*RH*T*D\ i$ | 2.42* | 2.51 |

8. Lines 262–264: I think this sentence should appear with the description of Plantower

> This has been moved alongside the description of the Plantower.
> Lines 139 to 140: The Plantower sensor reports estimated mass of particles with aerodynamic diameters <1 µm ($PM_1$), <2.5 µm $PM_{2.5}$, and <10 µm ($PM_{10}$).

9. If the authors want to stick with the conventional "methods" and "results" organization, the text on lines 255–256, 262–272, 276–277, 288–293, 299–300, 306, and 310–329 is technically methods. The models considered should be described in the methods section and then the amount of variance explained/comparison to prior studies should appear in the results section. If the authors wish to use some alternative organization, I think it would make sense to merge Section 2.3 with Section 3.3. (Model input consideration & Determining parameters and equations to use)

> We have heavily reorganized the paper so that it fits more consistently into the proposed alternate sections 2) Data Collection, 3) Quality Assurance, 4) Model Development, 5) Model Evaluation

TECHNICAL CORRECTIONS

Line 12: Please insert a comma after "sensors" and another after "(PM)".

> Line 12-13: PurpleAir sensors, which measure particulate matter (PM), are widely used by individuals, community groups, and other organizations including state and local air monitoring agencies.

Lines 45–50: This sentence contains five instances of "and" and zero commas, which made it difficult to follow. Please consider rephrasing or splitting it into multiple sentences.

> We have clarified this sentence:

> Lines 45-50: For optical particulate matter (PM) sensors, correction procedures are often needed due to both the changing optical properties of aerosols associated with both their physical and chemical characteristics (Levy Zamora et al., 2019;Tryner et al., 2019), and the influence of meteorological conditions including temperature and relative humidity (RH) (Jayaratne et al., 2018;Zheng et al., 2018). In addition, some air sensors have out of the box differences and low precision between sensors of the same model (Feenstra et al., 2019;Feinberg et al., 2018).

Line 53: Insert "(Plantower PMS5003)" after "particle sensors".
> Inserted Line 53: PurpleAir sensors are a PM sensor package consisting of two laser scattering particle sensors (Plantower PMS 5003),

Line 54: Specify "(Bosch BME280)" instead of "(BME280)".

> Inserted

> Line 54: pressure-temperature-humidity sensor (Bosch BME280),

Line 68: "location specific" should be "location-specific".

> Updated Line 74: location-specific

Line 71 and elsewhere: "U.S. wide" should be "U.S.-wide".

> We have updated this throughout.

Line 98: Please insert a comma after "sensors" and another after "B".

> Updated line 140: The PurpleAir sensor contains two Plantower PMS5003 sensors, labeled as channel A and B,

8. Lines 113–114: I suggest rewriting this sentence in past tense to be consistent with the rest of the paragraph (change "ensures" to "ensured" and "are" to "were").

> This sentence has been rewritten.

> Lines 271-272: This methodology ensured that the averages used were truly representative of daily averages reported by regulatory monitors.

9. Lines 135–136: This sentence was difficult to follow. Maybe rephrase as: "The Plantower sensor reports estimated mass of particles with aerodynamic diameters $< 1$ µm ($PM_1$), $< 2.5$ µm ($PM_{2.5}$), and $< 10$ µm ($PM_{10}$)."

10. Lines 145–146: I think "earliest data" should be "date".

> corrected

11. Line 148: "local time" should be "local standard time," correct?

> corrected

12. Lines 284–285: I assume "30 and 80%" refers to RH? Can you write "30% and 80% RH" to ensure this is clear?

> updated

13. Line 305: Insert a comma after "Lastly".

> corrected

14. Line 311: "corrections" should be "correction".

> corrected

15. Line 314: There are two periods at the end of this sentence.

> corrected

16. Line 406: The authors alternate between "quality assurance," "cleaning," and "quality control" throughout the manuscript. More consistent terminology would be preferable.

> We have simplified throughout to only use "quality assurance" when discussing exploring the quality of the data and "data cleaning" when discussing removing specific problematic data.

17. Line 420–421: Should this sentence read, "In this work, we also excluded three PurpleAir sensors because there was overall poor agreement between the A and B channels even after excluding individual data points."? If not, then maybe I do not understand what the authors are trying to say.

Yes, thank you for catching this mistake. This sentence does not appear in the final manuscript.

18. Line 562: Should there be an "and" between "correction methods" and "quality assurance methodology"?

Yes, thank you for pointing out this typo

**Lines 950-951:** Developing correction methods and data cleaning methodology for additional sensor types could further increase the amount of data available to communities, epidemiologists, decision makers, and others.

19. Line 559–561: Can the authors please revise this sentence? It's quite long and contains some redundant statements.

**Lines 946-949:** Most other sensor types do not contain duplicate $PM_{2.5}$ measurements which will make ensuring their data quality more challenging and more complex methods of data cleaning may be required, or similar data quality may not be possible.

20. Figure 1: I like how the authors use this figure to illustrate possible data quality issues, but it's really hard to see the data here. Can the authors make the whole figure larger, use smaller strip text so that the plot panels are larger, and use open circles as markers? Can the authors also either reorder the facets so that the PurpleAirs in CA, for example, appear in the order CA1, CA2, CA3, CA4, CA5, CA6, CA7, CA10, CA11, CA12, CA13, CA15, CA16, CA17, CA18, CA19? This could be accomplished by labeling "CA1" as "CA01" or by reordering the factor associated with these labels.

We have updated the figure also taking into account the other reviewer's suggestion to include correlation

[Figure]

Figure 2. Comparison of 24-hour averaged PM$_{2.5}$ data from the PurpleAir A and B channels. Excluded data (2.1%) are shown in red and represent data points where channels differed by more than 5 µg m$^{-3}$ and 61%. AK3, CA7, WA5 were excluded from further analysis. Pearson correlation (r) is shown on each plot.

21. Table S7: There is a broken reference at the end of the caption.

Updated.

---

## Referee Report (RR1)

Review for Development and Application of a United States wide correction for PM2.5 data collected with the PurpleAir sensor

Overall, the paper is well written and clear, and the quality of the revised manuscript improved a lot. I have minor comments that hopefully you will be able to address.

Line 90 you sate there were 14 states but then in line109 you wrote there were 16, this is a bit confusing how did you start with 14 but end up with 16.

Line 299 – you have double dote.

Line 401 no need to write full description of MBA or MAE again, you already define it in line 395

---

## Author Response (AR2)

Referee #1

Review for Development and Application of a United States wide correction for PM2.5 data collected with the PurpleAir sensor

Overall, the paper is well written and clear, and the quality of the revised manuscript improved a lot. I have minor comments that hopefully you will be able to address.

We thank the reviewer for their detailed final comments.

Line 90 you sate there were 14 states but then in line109 you wrote there were 16, this is a bit confusing how did you start with 14 but end up with 16.

This has been clarified:

Lines 91-92: From a download of all active AQS PM$_{2.5}$ sites and PurpleAir sensor locations on August 20, 2018, 42 unique sites were identified in 14 states (data from additional states were available from sensors sent out by EPA and privately operated sensors).

Line 299 – you have double dot.

This has been corrected.

Line 401 no need to write full description of MBA or MAE again, you already define it in line 395

Instead of linking the Figure number the whole caption was accidently linked so this has been removed.

Referee #2

GENERAL COMMENTS

I appreciate the authors' careful consideration of my previous comments! I've listed a few minor concerns below. All line numbers correspond to the revised manuscript.

We thank the reviewer for another thoughtful and detailed review.

SPECIFIC COMMENTS

1. Lines 199–204: "…occasionally, an extremely high temperature (i.e., 2147483447) or an extremely low temperature (i.e., -224 or -223) was reported…Temperature values above 540 C (1000 F) were excluded before calculating daily averages…" All temperatures between -224 and 1000 F were included? Why not use stricter criteria for excluding erroneous temperature measurements? The results of a quick web search suggest that temperatures in the U.S. have historically ranged from -80 to 135 F. I suspect that many other groups will adopt the quality assurance criteria proposed here, so I think it's worth scrutinizing those criteria.

This has been addressed in the text:

Lines 205-208: Temperature values above 540°C (1000°F) were excluded before calculating daily averages since temperature errors were extreme and easily detected above this level. After excluding these values reasonable 24-hr averaged temperature values were generated (min= -25°C, max=44°C). Future work may wish to apply a narrower range of acceptable temperature ranges, accounting for typical ambient conditions and the potential for increased heat build-up inside sensors (discussed further in section 4.1).

2. Lines 521–522: "Applying this correction to our dataset results in an underestimate of $PM_{2.5}$ by 3 ug/m3 (34%) on average…" Does this mean that MBE = -3 ug/m3? If so, can you please be more specific? This comment also applies to lines 529 and 534.

This has been clarified throughout this section.

Lines 526-527: Applying this correction to our dataset results in an underestimate of $PM_{2.5}$ by 3 $\mu g\ m^{-3}$ (MBE= -3 $\mu g\ m^{-3}$, 34%) on average with more scatter as quantified by the RMSE (LRAPA= 4 $\mu g\ m^{-3}$, US correction=3 $\mu g\ m^{-3}$).

Lines 533-536: Applying this correction to our dataset results in an overestimate of $PM_{2.5}$ of 4 $\mu g\ m^{-3}$ (MBE= 4 $\mu g\ m^{-3}$, 51%) with more scatter as quantified by the RMSE (AQ&U=6 $\mu g\ m^{-3}$, U.S. correction=3 $\mu g\ m^{-3}$).

Lines 539-541: Overall, applying this equation to our dataset results in a slight underestimate of $PM_{2.5}$ by 1 $\mu g\ m^{-3}$ (MBE= -1 $\mu g\ m^{-3}$, 12%) on average with a similar scatter as measured by RMSE (both Woodsmoke and US correction =3 $\mu g\ m^{-3}$).

3. Figures 5, 6, and S10: Were the corrected PurpleAir PM2.5 values plotted in these figures calculated using the coefficients shown in Equation 10 (which were obtained using the full dataset), or were they calculated using one of the withholding methods? I'm assuming the former, but it's not clear. Can the authors please clarify in the captions?

This has been clarified in the captions:

Figure 5. Error and ratio between corrected PurpleAir (PA) and FRM or FEM measurements are shown along with corrected PurpleAir $PM_{2.5}$ data (corrected using Eq. 10) as influenced by temperature, RH, and FRM or FEM $PM_{2.5}$ concentration. Colors indicate states, and black points indicate averages in 10 bins.

Figure 6. Scatterplot of the daily FEM or FRM $PM_{2.5}$ data with the PurpleAir data by U.S. region (see Figure 2) prior to any correction, after applying a linear correction, and after applying the final correction including RH. Data were corrected using the models built for the full dataset.

TECHNICAL CORRECTIONS

1. Line 105: "cited" should be "sited"

corrected

2. Lines 263–264: I think this should just say "(Figure 2, Table 1)" but the entire Figure 2 caption seems to have been inserted by mistake.

Thanks for catching this. It has been removed.

3. Lines 272–274: Now that the manuscript has been reorganized, the inaccuracies associated with PurpleAir temperature and relative humidity measurements have not been discussed yet. The authors could revise this sentence to something like "There were limited data above 80% RH as measured by the PurpleAir sensor, which, as discussed in the following section, is known to consistently report RH lower than ambient."

Thank you for this suggestion. We have updated the text

4. Line 302: "was" should be "were," since it refers to "equations"

changed

5. Line 325: Delete the comma after "sensors"

removed

6. Lines 401–403: The entire Figure 4 caption seems to have been inserted by mistake.

removed

7. Line 480: Delete "Lastly,". Sections 5.3.3 and 5.3.4 both start with "Lastly,".

We have removed lastly from the beginning of section 5.3.3

8. Line 583: "Norther" should be "Northern"

corrected

9. Table S9 caption: Should "(as summarized in Table A5)" say "(as summarized in Table S8)"?

Yes, updated

10. Figure S2: Do the authors want to remove the x-axis label ["xlab(NULL)"] or add one that isn't just their internal variable name?

Yes, we have removed the x-axis label.

Referee #3

It is unfortunate that the data set does not allow for a more in-depth investigation. It is also unfortunate that this was not addressed clearly earlier. My greatest complaint is – and has been since the first initial review – that this study, in its current form, does not make a significant scientific contribution to the field: there are already multiple studies investigating the effects of meteorological conditions on sensors, and many different correction equations have been presented. The notion that the correction equation presented here is similar to the ones presented previously makes this study less impactful in my opinion. The presented data processing technique is valuable, but it alone does not warrant for a full-length research article. Undoubtedly there will be readers who find this research useful, and there many published studies, which are of much worse scientific quality than this study; it is understandable if this paper gets published. However, to stay consistent, I will stand by my previous decision and will not recommend its publication. The field of low-cost sensor studies is not particularly novel anymore,

and research, which describes the sensors on a general level without diving deep into their metrological features, for example, will not suffice.

We thank the reviewer for their time spent reviewing our manuscript. No past work looked at a broadly applicable correction across the U.S. and so we feel that it strengthens our work that our correction is similar to corrections built for specific areas. Although low-cost sensors are widely used, there is still not a broad understanding of how to effectively and efficiently quality control and correct sensor data in order to make this data useful across large regions and so we feel that this paper fills a critical research gap.

We have added one sentence in response in the conclusion:

Lines 547-549: Although no previous work had attempted a broadly applicable correction, the correction developed in this paper is similar to those developed for specific locations or sources (i.e. smoke) strengthening the confidence that this correction is applicable across the U.S. This national evaluation suggested that any corrections that are not strictly local probably need to include RH or other environmental factors to represent the wide range of conditions that can occur in the US.